# Joint-Embedding Predictive Learning of Latent Market States in U.S. Equities

**Simon Mahns** [1]   **Randall Balestriero** [2]   **Mahmoud Assran** [3]

## Abstract

We investigate whether Joint-Embedding Predictive Architectures (JEPA) can learn useful representations of U.S. equity markets. We jointly train a permutation-invariant tokenizer that maps each trading day's unordered per-asset features to a fixed set of learned factor tokens, together with a temporal JEPA using masked prediction to obtain a compact daily market-state embedding. Our evaluations show that these embeddings are strongly associated with second-moment market structure (realized volatility, correlation concentration, effective factor dimensionality) and weakly associated with market direction. The embedding helps predict gradual recovery dynamics but not sudden stress onsets. Without any text supervision, latent regimes show statistically significant alignment with news-topic shifts.

## 1. Introduction

At daily frequency, equity markets exhibit variation on multiple axes. In this paper we emphasize two significant and orthogonal axes: direction and risk geometry. Directional index returns are noisy and are often driven by surprise news, while the cross-sectional *risk geometry*—the second-moment structure of returns, including volatility distribution, correlation matrix structure, and effective factor dimensionality—evolves more persistently and defines recognizable market regimes. This persistence reflects well-known empirical regularities such as volatility clustering and slow-moving correlation structure (Engle, 1982; Bollerslev, 1986). Our central question is: *with equity cross-sections alone, can a JEPA-style learning objective create a com-*

*pact market-state embedding that captures persistent cross-sectional risk geometry, volatility and correlation structure, while discarding market direction?*

We describe a "daily cross-section" as a set of per-equity feature vectors on day $t$, where many of the selected features are trailing-window summaries (e.g., 21-day realized volatility) using only information available up to $t$ (full feature set described in Table 7). In our proposed pipeline, we treat each daily cross-section as a set-valued object. This is enforced by our tokenizer, which compresses the daily universe of equities into a fixed set of $K$ learned "factor tokens." With these factor tokens, an encoder, trained with the JEPA framework with $L$-day token clips using masked prediction in embedding space (Section 2.4), produces a state representation $z_t$ designed to retain what is predictable from temporal and cross-sectional context while discarding idiosyncratic, hard-to-predict details (namely directional returns).

We focus on the JEPA objective because masked prediction in latent space naturally emphasizes shared, slowly varying structure across time and assets. In the equity setting, this creates strong pressure to encode persistent second-moment dynamics and little incentive to represent weakly predictable directional shocks.

**Scope**   This is a representation learning study. We learn embeddings and evaluate what they encode via probes, retrieval, geometry tests, and temporal dynamics. *We do not optimize for expected returns or claim implementable trading strategies.* Relative to established econometric and hand-crafted descriptors, the embedding is *complementary rather than uniformly superior.*

### 1.1. Related Work

**Market regimes and second-moment dynamics.**   Time variation in risk is a central part of financial econometrics. Volatility persistence is modeled by ARCH/GARCH (Engle, 1982; Bollerslev, 1987) and realized-volatility / HAR-style summaries (Andersen et al., 2003; Corsi, 2009), while evolving dependence is often captured via dynamic correlation models such as DCC (Engle, 2002). Discrete regime formulations are commonly studied with Markov switching (Hamilton, 1989), and empirical "market states" have been

---

[1]Whiting School of Engineering, Johns Hopkins University, Baltimore, MD, United States [2]Department of Computer Science, Brown University, Providence, RI, United States [3]Mila – Quebec Artificial Intelligence Institute, Montreal, Quebec, Canada. Correspondence to: Simon Mahns <ak.simonm@gmail.com>, Randall Balestriero <randall_balestriero@brown.edu>, Mahmoud Assran <massran@meta.com>.

*Proceedings of the $43^{rd}$ International Conference on Machine Learning*, Seoul, South Korea. PMLR 306, 2026. Copyright 2026 by the author(s).

defined by similarity and clustering of correlation matrices. *In contrast*, we do not fit an explicit regime model to hand-designed second-moment statistics; we learn a compact *continuous* state embedding directly from daily cross-sections.

**Factor structure and latent representations in asset pricing.** Factor models compress the cross-section into a small number of systematic sources of variation, from CAPM/APT to multifactor models (Sharpe, 1964; Ross, 1976; Fama & French, 1993). More recent work allows time-varying exposures driven by characteristics (e.g., IPCA) (Kelly et al., 2019) and learns latent factor structure with neural networks under return supervision (Gu et al., 2021). *Our objective differs* as we are not optimizing for expected-return prediction or pricing errors. Instead, we learn day-level market-state representations that emphasize persistent cross-sectional *risk geometry* (volatility and correlation structure) while down-weighting directional shocks.

**Self-supervised representation learning with latent prediction.** JEPAs learn representations by predicting in embedding space rather than reconstructing inputs (LeCun & Courant, 2022; Assran et al., 2023), with extensions to spatiotemporal settings in Video JEPA 1 and 2 (Bardes et al., 2024; Assran et al., 2025). Related masked-teacher objectives include data2vec (Baevski et al., 2022), and recent work explores JEPA-style training for time series (Ennadir et al., 2025; Verdenius et al., 2024; Dutta et al., 2025). *We adapt this line of work to daily equity cross-sections* by operating on a time $\times$ token grid of learned factor tokens, using lag-aware positional encodings and a multi-stride curriculum.

**Set-structured tokenization and latent bottlenecks.** Daily equity universes are naturally sets (arbitrary ordering; missing assets). Permutation-invariant set models include Deep Sets (Zaheer et al., 2017) and attention-based Set Transformers (Lee et al., 2019); latent bottleneck designs such as the Perceiver distill large inputs into a compact latent array (Jaegle et al., 2021; 2022). *Building on this work*, we tokenize each day's unordered cross-section into $K$ learned factor tokens, enabling scalable temporal modeling without imposing a panel order and handling missing assets via masking.

## 2. Self-Supervised Learning of U.S. Equities with JEPA

### 2.1. Notation and Setup

We proceed with three steps:

1. **Data**: Creation of the data universe.

2. **Equity tokenization**: Map the daily cross-section $\mathbf{X}_t$

into $K$ factor tokens, $\mathbf{T}_t$, capturing contemporaneous cross-asset interactions.

3. **Temporal market representation learning**: Across a temporal context of $L$, where each day is represented as $\mathbf{T}_t$, learn a risk geometry focused representation of the market state directly in the embedding space.

### 2.2. Data

We use daily aggregates from Massive.com for U.S. equities traded on NYSE, NASDAQ, and NYSE American. The universe consists of 512 equities selected by average dollar volume computed *on the training period only*, freezing the selection before validation and test data are observed. This avoids look-ahead in universe construction but introduces a liquidity selection bias: the top-ADV filter excludes less liquid securities, limiting conclusions to highly traded equities.

All prices are split-adjusted using point-in-time adjustment factors; dividend adjustments are not applied (the data vendor does not provide dividend-adjusted prices), so all returns are price returns. On each trading day, we observe between 480 and 512 active equities (median 508); assets missing on a given day receive null features and are excluded via a validity mask. Temporal splits are strict: training covers 2018–2022, validation is 2023, and the test set is 2024–2025.

Feature warmup (126-day lookback) and clip length ($L{=}21$) consume the first $\sim$150 days; the effective evaluation period is May 2018 through October 2025 ($T_{\text{eval}}{=}1{,}864$ trading days). Feature normalization statistics are computed on training data only. Full universe and feature definitions are in Appendix A.[1]

**Feature groups.** The $F{=}28$ per-equity features cover returns and horizon structure, realized and EWMA volatility, OHLC candle geometry, volume and liquidity, and market-relative quantities; full definitions are in Table 7.

### 2.3. Permutation-Invariant Cross-Sectional Tokenizer

We treat each daily equity cross-section as an unordered set $\mathbf{X}_t = \{\mathbf{x}_{t,i}\}_{i=1}^{N_t}$ rather than a fixed-order panel. To obtain a fixed-shape representation suitable for temporal modeling, we map $\mathbf{X}_t$ to $K$ learned "factor" tokens $\mathbf{T}_t \in \mathbb{R}^{K \times d}$ using a Set Transformer front-end (Lee et al., 2019). We use three building blocks from that framework: a *self-attention block* (SAB), which is a residual multihead self-attention layer

---

[1]Throughout, we distinguish: $T_{\text{raw}}{=}2{,}010$ (raw trading days), $T_{\text{feat}}{=}1{,}884$ (after feature warmup), $T_{\text{eval}}{=}1{,}864$ (after clip burn-in). Test split contains 451 raw days; effective test sizes vary by evaluation due to warmup and purging requirements (e.g., $n{=}365$ for geometry probes requiring non-overlapping windows).

over a set; an *inducing-point self-attention block* (ISAB), which replaces $O(N^2)$ pairwise attention with attention through $m$ learned inducing points at cost $O(Nm)$; and *pooling by multihead attention* (PMA$_K$), which uses $K$ learned seed vectors to attend over the set and pool it to exactly $K$ output tokens. Each equity is first embedded independently by a row-wise MLP $\psi : \mathbb{R}^F \to \mathbb{R}^d$, cross-asset interaction is modeled by $B_{\mathrm{ISAB}}$ ISAB blocks, and the set is then pooled to $K$ tokens by PMA, followed by a single SAB over the pooled tokens:

$$\mathbf{T}_t = \tau_\omega(\mathbf{X}_t) = \mathrm{SAB}\Big[\mathrm{PMA}_K\Big(\mathrm{ISAB}_m^{(B_{\mathrm{ISAB}})}(\psi(\mathbf{X}_t))\Big)\Big].$$
$$(1)$$

This mapping produces a time-invariant $K \times d$ token set; missing equities are handled via attention masks. Unless otherwise noted, we use $K{=}24$, $d{=}128$, $m{=}32$, and $B_{\mathrm{ISAB}}{=}2$. Full implementation details are in Appendix B.1.

**Design rationale.** Set modeling provides permutation invariance (essential since equity ordering is arbitrary) and graceful handling of missing assets (common in daily panels). Inducing-point attention enables scaling to $N_{\max}{=}512$ equities without quadratic cost. The $K$ factor tokens serve as learned "market factors," compressing the cross-section into a fixed-shape summary suitable for temporal JEPA.

## 2.4. Temporal Market Representation Learning

In the preceding step we learned the *contemporaneous* relationships between equities at time $t$, the objective in this step is to learn regime-level dynamics across time/market states (risk geometry) directly in representation space. In practice, this means taking a temporal context window of length $L$, where each day is separated by stride $s$ days and represented by $\mathbf{T}_t$, and producing a market-state embedding $\mathbf{z}_t$ for each day. We denote the stride-$s$ clip ending at day $t$ as $\mathbf{H}_t^{(s)} = [\mathbf{T}_{t-(L-1)s}, \ldots, \mathbf{T}_t]$, an $L \times K$ time–token grid. For the transformer implementation, we flatten this grid into a length-$LK$ sequence and add a positional embedding $\mathrm{pos}(\ell, k; s)$ to each token.

### 2.4.1. ARCHITECTURE: CONTEXT ENCODER, PREDICTOR, AND TARGET ENCODER

We adopt the standard JEPA meta-architecture:

- a *context encoder* $f_\theta$ that processes visible tokens (each augmented with $\mathrm{pos}(\ell, k; s)$) and outputs contextualized embeddings;

- a *predictor* $p_\phi$ that takes the encoded visible tokens together with learnable mask tokens at masked positions and outputs predictions;

- a *target encoder* $f_{\bar\theta}$, an EMA copy of $f_\theta$, used to generate stable regression targets.

We define the market state $\mathbf{z}_t \in \mathbb{R}^d$ as the mean-pooled representation over the $K$ factor tokens at the final timestep of the clip. Unless otherwise noted, we use stride $s = 1$.

### 2.4.2. MASKED SPATIOTEMPORAL PREDICTION

Rather than encoding sequence index, we encode the *trading-day lag* (time-to-present):

$$\Delta(\ell) = (L - 1 - \ell) \times s \quad \text{(in trading days)},$$

using Fourier features with logarithmically spaced frequencies (Tancik et al., 2020). The Fourier basis provides a smooth multi-frequency encoding that allows the model to extrapolate across different temporal horizons. With stride $s$, $\Delta(\ell)$ changes by $s$ trading days per step. The full positional embedding is

$$\mathrm{pos}(\ell, k; s) = e_{\mathrm{lag}}(\Delta(\ell)) + e_{\mathrm{slot}}(k) \qquad (2)$$

so that different strides are distinguished through $\Delta(\ell)$. We define the *effective horizon* as $H(s) = (L - 1)s$ trading days; with $L = 21$, our stride curriculum $s \in \{1, 3, 7, 21\}$ yields horizons $H(s) \in \{20, 60, 140, 420\}$ trading days. Specific stride curriculum details are outlined in Table 9.

Masking is applied over the $L \times K$ time–token grid. For each clip, we sample a masked index set $\mathcal{M} \subseteq \{0, \ldots, L-1\} \times \{1, \ldots, K\}$; the loss is computed only over masked factor tokens. In our implementation, masks are contiguous in time but random over factor slots, with a mixture of: (i) time-span $\times$ random-slot masks, (ii) full-day masks (all slots on selected days), and (iii) full-slot masks (selected slots across all time). The complement $\mathcal{V}$ denotes visible indices. Masked positions are represented by learnable mask tokens augmented with positional information.

### 2.4.3. OBJECTIVE

Let $\mathbf{Y} = f_{\bar\theta}(\mathbf{H}_t^{(s)}) \in \mathbb{R}^{L \times K \times d}$ denote the target embeddings, and let $\mathrm{sg}(\cdot)$ denote stop-gradient. The context encoder produces

$$\mathbf{Z}_\mathcal{V} = f_\theta\Big(\mathbf{H}_t^{(s)}\big|_\mathcal{V}\Big),$$

and the predictor produces $\widehat{\mathbf{Y}}_\mathcal{M} = p_\phi(\mathbf{Z}_\mathcal{V}, \mathcal{M})$. We minimize an $\ell_1$ regression loss on masked tokens:

$$\mathcal{L}_{\mathrm{JEPA}} = \frac{1}{|\mathcal{M}|} \sum_{u \in \mathcal{M}} \Big\|\widehat{\mathbf{Y}}_u - \mathrm{sg}(\mathbf{Y}_u)\Big\|_1. \qquad (3)$$

We apply LayerNorm to target embeddings before computing the loss. The tokenizer $\tau_\omega$, context encoder $f_\theta$, and predictor $p_\phi$ are trained jointly via $\mathcal{L}_{\mathrm{JEPA}}$; the target encoder

*Table 1.* **Evaluation suite.** Seven evaluations to investigate what the market-state embedding encodes. Metric and financial-term definitions are in Appendix F.

| Evaluation | Question | Metric |
|---|---|---|
| Linear probes | Can geometry scalars be decoded from $\mathbf{z}_t$? | Ridge $R^2$ |
| kNN retrieval | Do neighbors share market descriptors? | Spearman $\rho$ ($\pm21$d excl.) |
| Distance alignment | Do latent distances match geometry distances? | Mantel $\rho$ (gap $\geq63$d) |
| Neighborhood geometry | Are neighbors closer in geometry than random? | % improvement |
| Transition prediction | Does $\mathbf{z}_t$ add predictive value? | $\Delta$AUC, $\Delta$PR-AUC |
| Dynamics | Do displacements detect stress? | Enrichment OR |
| Cross-modal | Do neighbors share news? | $\Delta(W)$, test |

$f_{\bar{\theta}}$ is an EMA copy of $f_\theta$ (the tokenizer $\tau_\omega$ is shared, and not included in the EMA process), with $\bar{\theta} \leftarrow \beta\bar{\theta} + (1-\beta)\theta$ after each step.

## 2.5. Implementation Summary

The context encoder and predictor are 6-layer transformers with 4 attention heads. EMA momentum $\beta$ is scheduled from 0.996 to 0.9999 over training. Full hyperparameters are in Appendix C.1; stride curriculum details are in Table 9.

## 3. Evaluation and Investigation

Unless specified otherwise, the following investigation is done on market-state embeddings produced by a tokenizer and frozen JEPA encoder that were trained jointly (see Appendix B). We extract embeddings matching the training configuration for context window ($L = 21$) over the period 2018–2025. For each day $t$, we encode the sequence $[\mathbf{X}_{t-20}, \ldots, \mathbf{X}_t]$ ($s = 1$) and extract the market-state embedding $\mathbf{z}_t$.

## 3.1. Visualizing the Embedding Space

Before presenting quantitative metrics, we visualize the learned representation space. Figure 1 shows Representation Similarity Matrices (RSMs) comparing tokenizer-only and JEPA-refined latents. Both recover regime-level blocks and long-range temporal correspondences, but JEPA reduces the concentration of extreme similarities, spreading mass toward intermediate values. The visible block structure

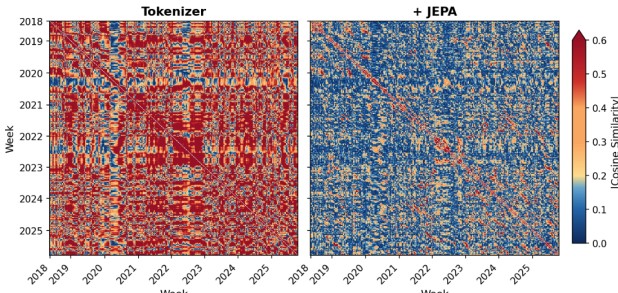

*Figure 1.* **Representation Similarity Matrices (RSMs) for tokenizer-only and JEPA-refined latents.** We visualize the magnitude of pairwise cosine similarity between weekly-averaged market-state embeddings. Both representations exhibit the same coarse geometric structure, including regime-level blocks and temporal continuity. JEPA reduces the concentration of extreme similarities, redistributing mass toward intermediate values and increasing the proportion of differentiated state pairs.

and smooth temporal evolution motivate the quantitative evaluations that follow.

## 3.2. Neighborhood Semantics: Risk Structure, Not Direction

We assess the semantic content of the learned market states via $k$-nearest-neighbor (kNN) retrieval in embedding space. For each query day $t$, we retrieve its $k = 20$ nearest days by cosine similarity, excluding matches within a $\pm21$-day window to mitigate temporal leakage. We then evaluate whether embedding neighborhoods preserve interpretable market descriptors by reporting Spearman rank correlations (Spearman, 1904) between query-day descriptors and their neighbor-based estimates. Figure 2 shows strong alignment with volatility and correlation geometry (e.g., $\rho = 0.78$ for RV21 and $\rho = 0.68$ for PC1 variance share), but weak association with market direction ($\rho = 0.11$ for market return).

## 3.3. Geometry Alignment Beyond Temporal Proximity

**Geometry scalars.** We summarize the daily cross-sectional correlation structure with three scalar metrics computed from the trailing 21-day return correlation matrix:

- corr_mean: mean pairwise correlation across all asset pairs;

- pc1_share: fraction of total variance explained by the first principal component;

- eff_rank: effective rank, defined as $\exp(H)$ where $H$ is the entropy of the normalized eigenvalue spectrum.

These capture complementary aspects of cross-sectional

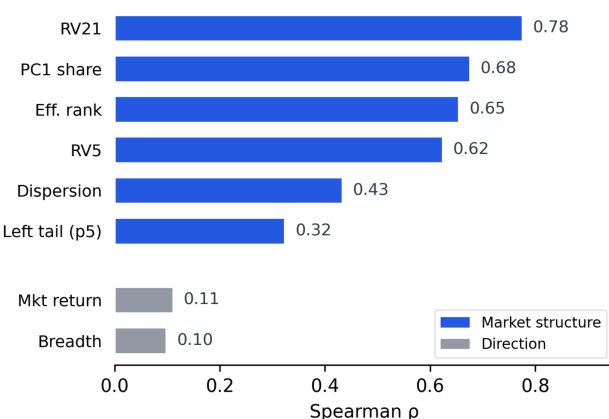

*Figure 2.* **kNN retrieval evaluation.** Spearman rank correlation between query-day descriptors and neighbor-based descriptor estimates ($k$=20, $\pm$21-day exclusion). Blue: market-structure measures; gray: directional indicators. RV21/RV5: realized volatility at 21/5 day; PC1 share: variance explained by the first principal component; Eff. rank: effective rank of the return covariance; Left tail (p5): 5th percentile of cross-sectional returns.

geometry: `corr_mean` measures average comovement, `pc1_share` measures factor concentration, and `eff_rank` measures the effective dimensionality of the return distribution. High-stress regimes typically exhibit elevated `corr_mean` and `pc1_share` with depressed `eff_rank`; recovery is characterized by the reverse pattern.

Table 2 and Fig. 3 summarize two evaluation questions about what the learned embedding represents.

**Does JEPA encode current geometry?** As a prerequisite, we verify that a frozen linear probe can decode contemporaneous geometry levels from $\mathbf{z}_t$. Ridge regression on held-out data yields $R^2 = 0.42$ for `corr_mean`, $R^2 = 0.45$ for `pc1_share`, and $R^2 = 0.36$ for `eff_rank` (all $p < 10^{-20}$). This confirms the representation encodes cross-sectional geometry in a linearly accessible form.

### 3.3.1. GEOMETRY FIDELITY

We test whether *distances in latent space* correspond to *distances in a market geometry*. We use a Mantel-style test (Mantel, 1967), which correlates two distance matrices: latent distances $D_{ij}^{\text{latent}} = \|\mathbf{z}_i - \mathbf{z}_j\|$ and geometry distances $D_{ij}^{\text{geom}}$ computed as the Frobenius norm between trailing correlation matrices. We report Spearman $\rho$ between the vectorized upper triangles of these distance matrices.

Using pairs of days separated by at least 63 trading days reduces the time-adjacency confound: a time-only embedding collapses ($\rho = -0.10$), while JEPA remains positively aligned ($\rho = 0.35$, 95% CI excludes zero). A volatility-only baseline is also strong ($\rho = 0.44$), indicating that

correlation geometry is substantially explained by volatility regimes—this is expected, as volatility is a dominant factor in correlation dynamics. A spectrum-derived oracle achieves $\rho = 0.68$, serving as a ceiling.

The strength of the volatility baseline is not a failure of JEPA. Rather, it reflects a well-known empirical regularity: volatility explains much of the cross-sectional correlation structure. The contribution of the JEPA embedding is the *additional* structure it captures beyond volatility, as evidenced by incremental recovery prediction gains (Section 3.3) and cross-modal news alignment (Section 3.5). kNN retrieval with temporal exclusion shows JEPA neighbors are +6.7% closer in geometry distance than random neighbors. To rule out that the embedding is merely a passthrough for the explicit volatility input channels, we retrain end-to-end with all eight removed: the Mantel $\rho$ degrades ($0.35 \rightarrow 0.32$) and the recovery AUC gain roughly halves ($+0.24 \rightarrow +0.14$) but both persist (Appendix D.1).

### 3.3.2. PREDICTIVE UTILITY

We test whether the market-state embedding adds forward-looking information beyond simple geometry scalars. For each task, we compare a logistic regression on geometry scalars to the same model augmented with the market-state embedding.

**Task definition.** We predict whether a geometry metric will increase over the next 21 trading days. Past and future windows are non-overlapping (no shared days); a day is labeled positive if its change exceeds the 90th percentile of training-set changes. Splits include a 21-day purge buffer to prevent label leakage.

The asymmetry in Fig. 3 is notable: JEPA provides large gains for *diversification recovery* (effective-rank increases: $\Delta\text{AUC} = +0.24$, $\Delta\text{PR-AUC} = +0.37$), but provides little or no benefit for stress-onset prediction (`corr_mean`$\uparrow$) and *hurts* generic large-magnitude transition detection. This follows from the inductive bias described in Section 1: recovery is gradual and predictable; stress onsets are abrupt surprises.

**Comparison to domain baselines.** On the same task, split, and protocol, we compare against three domain baselines: (i) HAR, a heterogeneous autoregressive model on geometry levels (Corsi, 2009); (ii) a Gaussian hidden Markov model on market observables; and (iii) a hand-crafted baseline of six market-state descriptors (`logRV21`, dispersion, breadth, left tail, `pc1_share`, `eff_rank`) with daily/weekly/monthly persistence summaries (18 features total). JEPA's only clear gain is on diversification recovery (`recovery_eff_rank_UP`: ROC-AUC $0.690 \rightarrow 0.800$, AP $0.139 \rightarrow 0.324$); on the other transition tasks it is small

| Method | Mantel $\rho$ (gap $\geq$ 63d) | kNN geom. $\Delta$ (%) |
|---|---|---|
| Time baseline | -0.10 | – |
| Vol-only | 0.44 | – |
| JEPA | 0.35 (0.16, 0.54) | +6.7 |
| Spec. (oracle) | 0.68 | +11.0 |

*Table 2.* **Geometry alignment of embedding distances.** Mantel-style Spearman correlation between embedding distance and correlation-geometry distance with a minimum time separation of 63 trading days, and kNN retrieval improvement relative to random neighbors ($k$=10, $\pm$21d exclusion). JEPA remains positively aligned after removing temporal proximity, while a time-only baseline collapses. Spectrum features serve as an oracle ceiling.

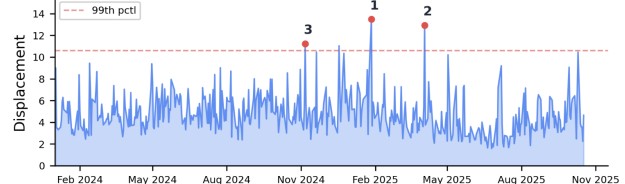
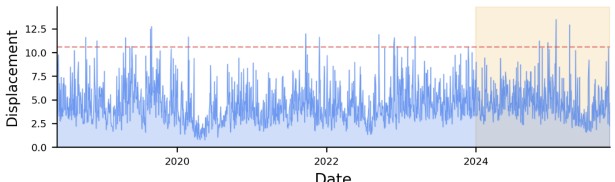

*Figure 4.* **Embedding space dynamics.** Daily displacement magnitude $\delta_t = \|\mathbf{z}_t - \mathbf{z}_{t-1}\|_2$. The dashed line indicates the 99th percentile threshold. Bottom: full evaluation period. Top: days after Jan 1, 2024. The three largest displacements in this period (selected post-hoc as top-3 by magnitude) coincide with widely reported market events: (1) January 27, 2025: technology sector selloff following DeepSeek AI announcement,[1] (2) April 3, 2025: tariff-driven selloff,[2] and (3) November 6, 2024: post-election market rally.[3]

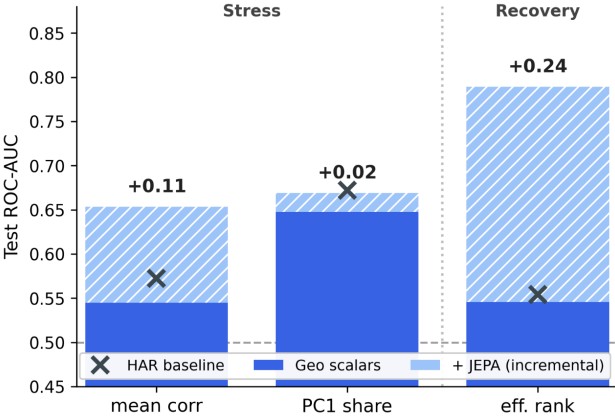

*Figure 3.* **Incremental predictive value of JEPA beyond geometry scalars.** Test AUC for predicting geometry transitions: stress-onset (mean correlation ↑, PC1 share ↑) and recovery (effective rank ↑). Solid bars: geometry scalars only. Hatched: geometry + market-state embedding. X marks: HAR baseline (heterogeneous AR on geometry levels). JEPA provides large gains for recovery (+0.24) but minimal improvement for stress prediction (+0.11, +0.02).

*Table 3.* Embedding displacement on event versus non-event days. Market stress events produce large and significant embedding shifts, while scheduled macro events do not. $N$ = event days; Effect = weekday-adjusted displacement difference; $d$ = Cohen's $d$; $p$ = significance level.

| Category | $N$ | Effect | $d$ | $p$ |
|---|---|---|---|---|
| FOMC | 66 | 0.12 | 0.06 | .63 |
| CPI Release | 89 | 0.06 | 0.03 | .79 |
| Crash Days | 30 | 1.78 | 0.92 | <.001 |
| VIX Spikes | 46 | 2.80 | 1.47 | <.001 |
| Market Stress | 59 | 2.26 | 1.18 | <.001 |
| All Systematic | 203 | 0.73 | 0.37 | <.001 |

or negative (Appendix D.6, Table 17).

### 3.4. Dynamics: Embedding Displacement as a Stress Indicator

We now shift from static geometry to temporal dynamics. Let $\delta_t := \|\mathbf{z}_t - \mathbf{z}_{t-1}\|_2$ denote the displacement between consecutive trading days (Fig. 4). Large values of $\delta_t$ indicate the learned cross-sectional configuration has shifted in representation space. The displacement series clusters in time (elevated during early 2020 and late 2022, subdued during mid-2021), consistent with regime-dependent dynamics rather than measurement noise.

---

[1]Reuters, "DeepSeek sparks AI stock selloff; Nvidia posts record market-cap loss," Jan. 27, 2025.

[2]The Guardian, " 'Liberation day': what are tariffs and why do they matter?," Apr. 3, 2025.

[3]Reuters, "Stocks surge to record highs as Trump returns to presidency," Nov. 6, 2024.

Table 3 compares mean displacement magnitudes on event versus non-event days. Market stress events (crash days, VIX spikes) produce significantly elevated displacements ($d > 0.9$, $p < 0.001$), while scheduled macro releases (FOMC, CPI) show no significant effect. Enrichment analysis confirms this asymmetry: top-1% displacements are strongly over-represented on stress days (OR= 32.3, Fisher $p<10^{-4}$) but not on announcement days (OR= 1.3, $p$=0.47); see Appendix D.3 for full enrichment statistics. This indicates that $\delta_t$ functions as a label-free indicator of realized market stress, the embedding moves when the cross-sectional configuration changes structurally, not merely when the calendar says an announcement occurred.

**Comparison to standard risk measures.** We test whether $\delta_t$ adds information beyond standard risk signals on the held-out split (test 2024–2025): |SPX return|, 5d rolling SPX volatility, 63d SPX drawdown, VIX level, |$\Delta$VIX|,

*Table 4.* **Latent regime news profiles.** Regimes are obtained by $k$-means clustering ($k=8$) on JEPA market-state embeddings; $n$ denotes the number of trading days assigned to each regime. Each cell reports the $z$-score of the family's mean share within that regime relative to the global mean across all days. Positive values indicate over-representation; negative values indicate under-representation. Bold: $|z| > 0.15$; entries with $|z| < 0.05$ shown as '–'. Abbreviations: NAT = NATURAL, MADE = MANMADE, UNR = UNREST.

| | R1 330 | R2 315 | R3 246 | R4 245 | R5 233 | R6 192 | R7 192 | R8 111 |
|---|---|---|---|---|---|---|---|---|
| ECON | **-0.25** | 0.05 | – | **0.15** | – | **0.19** | -0.06 | 0.09 |
| EMERG | **-0.19** | – | – | -0.06 | 0.13 | – | **-0.15** | **0.61** |
| NAT | -0.10 | – | **0.16** | 0.13 | 0.11 | **-0.24** | 0.10 | **-0.37** |
| MADE | **-0.23** | 0.07 | 0.10 | 0.09 | 0.07 | – | -0.03 | -0.06 |
| MED | **-0.33** | 0.15 | 0.06 | 0.03 | 0.06 | 0.15 | **-0.18** | **0.27** |
| UNR | **-0.26** | 0.07 | 0.06 | 0.12 | 0.05 | **0.17** | -0.06 | -0.12 |
| GOV | **-0.29** | 0.10 | 0.12 | -0.05 | 0.06 | 0.13 | -0.08 | **0.25** |
| ENV | **-0.22** | 0.06 | 0.05 | 0.08 | 0.07 | 0.13 | -0.04 | -0.09 |

HY spread level and change, and a cross-sectional liquidity proxy (median log dollar volume). As a standalone signal on top-1% Crash+VIX enrichment, $\delta_t$ matches the strongest individual signals (JEPA 14.96×, |SPX return| 14.96×, VIX 13.30×, drawdown 11.63×, liquidity 4.99×). Added to the full risk bundle, the gain is small and the paired-bootstrap CI covers zero on both Crash+VIX and All-Systematic (Appendix D.4, Table 14).

## 3.5. External Validity: Cross-Modal Alignment with News Topic Composition

### 3.5.1. SETUP

To assess whether the learned latent space captures economically meaningful structure, we design a cross-modal retrieval task linking latent geometry to external news data. We use the GDELT Global Knowledge Graph (GDELT Project, 2013), which provides daily counts of news articles tagged with hierarchical theme codes (e.g., ECON_BANKRUPTCY, ENV_CLIMATECHANGE). Over 1,250 unique themes appear in our evaluation period; we aggregate these into eight broad families by the first prefix before the underscore delimiter: ECON (491 themes), NATURAL (186), MANMADE (118), ENV (22), UNREST (18), GOV (11), EMERG (8), and MED (7). For each trading day, we construct a normalized news vector ($\log(1 + \text{count})$ weighting, L2-normalized) representing its news composition across these families. Full details on GDELT data extraction and theme aggregation are provided in Appendix E.

**Evaluation protocol.** For each query day $t$, we retrieve its $k = 5$ nearest neighbors in latent space, excluding days within a temporal window of $\pm W$ trading days to prevent autocorrelation from trivially inflating similarity. We then compute the cosine similarity between the query day's news

vector and the average news vector of its latent neighbors. Our key metric is

$$\Delta(W) = \text{Sim(latent neighbors)} \\ - \text{Sim(shuffled neighbors)}. \tag{4}$$

where the shuffled baseline is computed by circularly shifting the latent time series by a random offset larger than $W$ (repeated across trials), preserving the latent trajectory's temporal structure while breaking alignment to news. A positive $\Delta(W)$ indicates that proximity in latent space corresponds to shared news environments beyond what temporal proximity alone would predict.

### 3.5.2. RESULTS

We observe statistically significant alignment for large exclusion windows. On the train+validation period (2018–2023), $\Delta(W)$ is significantly positive for $W \geq 126$ (95% CI excludes zero), with $W = 63$ marginal (CI boundary near zero). The effect reaches $\Delta(252) = 0.0069$ (95% CI: [0.0015, 0.0131]). On the held-out test period (2024–2025), significance emerges earlier ($W \geq 21$): $\Delta(252) = 0.0077$ (95% CI: [0.0015, 0.0140]); see Table 15 for full curves.

**Ruling out temporal and seasonal confounds.** Because neighbors are constrained to be at least $W$ trading days away, $\Delta(W)$ cannot be driven by short-horizon autocorrelation in markets or news: for $W \in \{63, 126, 252\}$, nearest neighbors are separated by months to a year. A remaining concern is that $\Delta(W)$ could instead arise from calendar seasonality (e.g., recurring time-of-year news patterns) or from market stress proxies such as volatility. To address this, we rerun the identical retrieval evaluation using (i) a calendar seasonality representation (day-of-week and month indicators) and (ii) a volatility-only representation (log realized volatility). At $W = 63$ on the train+validation period, JEPA's $\Delta(63) = 0.0053$ exceeds the vol-only baseline (0.0024); a JEPA model retrained with all eight explicit volatility input channels removed (Appendix D.1) retains comparable alignment ($\Delta(63) = 0.0080$). The seasonality baseline never achieves significance. Full curves and confidence intervals are in Appendix D.5.

Table 4 further decomposes this alignment by clustering the latent space into eight regimes and examining news-family $z$-scores. Several patterns emerge: **R1** (330 days) is uniformly underrepresented across all news families, consistent with a "quiet" or low-salience market state; **R8** (111 days) shows strong overrepresentation of EMERG (+0.61), MED (+0.27), and GOV (+0.25), suggesting crisis or pandemic-related periods; **R3** exhibits elevated NAT (+0.16), indicative of natural-disaster-heavy news environments; and **R6** shows heightened ECON (+0.19) and UNR (+0.17) with suppressed NAT (−0.24), suggesting

*Table 5.* **JEPA vs MAE ablation.** kNN retrieval $R^2$ ($k$=20, $\pm$21d exclusion). Structure descriptors measure volatility and correlation geometry; direction descriptors measure market returns. $\Delta$ = JEPA − MAE. JEPA outperforms MAE on all structure descriptors, with the largest gains on correlation geometry (pc1_share, eff_rank). Both models are weak on direction, confirming that neither objective learns return prediction.

| | Descriptor | JEPA | MAE | $\Delta$ |
|---|---|---|---|---|
| Structure | RV21 (log) | **0.62** | 0.54 | +0.08 |
| | RV5 (log) | **0.41** | 0.32 | +0.09 |
| | PC1 share | **0.29** | 0.12 | +0.17 |
| | Eff. rank | **0.24** | 0.15 | +0.09 |
| | Dispersion | **0.21** | 0.15 | +0.06 |
| | Left tail | **0.27** | 0.23 | +0.04 |
| Dir. | Mkt. return | 0.07 | 0.05 | +0.02 |
| | Breadth | 0.02 | 0.02 | 0.00 |
| | *Mean (structure)* | **0.34** | 0.25 | +0.09 |
| | *Mean (direction)* | 0.05 | 0.04 | +0.01 |

*Table 6.* **Masking strategy ablation.** Spearman $\rho$ for structure descriptors (mean of RV21, RV5, PC1, eff. rank, dispersion, left tail) and direction descriptors (mean of mkt. return, breadth). Gap = structure − direction. Both models achieve similar structure encoding; the key difference is direction leakage.

| Masking | Structure $\rho$ | Direction $\rho$ | Gap |
|---|---|---|---|
| Structured (ours) | 0.56 | 0.14 | +0.42 |
| Random i.i.d. | 0.55 | 0.19 | +0.36 |

This ablation supports the paper's central claim: JEPA's latent prediction objective specifically encourages representations organized by persistent risk geometry rather than by input reconstruction fidelity. MAE, which optimizes reconstruction loss, captures some structure but allocates representational capacity more uniformly across features, including those less relevant to regime identification.

The same gap appears on the transition tasks: JEPA's recovery gain is roughly $3\times$ MAE's, and MAE's marginal contribution is not statistically supported across the four tasks (Appendix D.6, Table 17).

### 4.2. Masking Strategy

We ablate the masking strategy by comparing structured masking (contiguous time spans, used in all main experiments) against random i.i.d. masking. Both models use identical architecture and training; they differ only in how masked positions are selected. Table 6 reports kNN retrieval Spearman $\rho$ ($k$=20, $\pm$21d exclusion).

Random masking achieves comparable structure encoding ($\rho = 0.55$ vs. $0.56$) but encodes notably more directional information ($\rho = 0.19$ vs. $0.14$). This demonstrates that masking strategy acts as an inductive bias controlling what the representation emphasizes. Structured masking, which removes contiguous time spans, forces the model to predict from temporally distant context, where persistent cross-sectional patterns provide signal but short-horizon returns do not. Random masking permits prediction from nearby tokens that share similar returns, allowing directional information to enter the representation. For applications emphasizing regime identification over return prediction, structured masking is the appropriate choice.

## 5. Conclusion

Our evaluation suite confirms that the proposed pipeline learns embeddings encoding risk geometry (volatility, correlation structure, effective dimensionality) while largely ignoring market direction—answering the question posed in Section 1.

Compared with traditional econometric and handcrafted baselines, the embedding is *complementary rather than uni-*

economic-stress regimes distinct from natural disasters.

Importantly, these regime–news associations emerge without any news supervision during training; the latent space is learned purely from equity cross-sections. The news profiles serve only as a post-hoc lens. The held-out replication (2024–2025) suggests the associations generalize beyond the training window.

## 4. Ablations

### 4.1. JEPA vs MAE Baseline

We compare JEPA's latent prediction objective against a masked autoencoder (MAE) baseline that reconstructs token features rather than predicting in embedding space (He et al., 2021). Both models share identical architecture (tokenizer, encoder depth, embedding dimension) and training protocol; they differ only in objective. Table 5 reports kNN retrieval $R^2$ ($k$=20, $\pm$21d exclusion) for structure and direction descriptors; we use $R^2$ here rather than Spearman $\rho$ (as in Section 3.2) because coefficient of determination is more common for ablation comparisons. Extended results including baseline comparisons and sensitivity to $k$ appear in Appendix D.7.

JEPA outperforms MAE on all six structure descriptors, with the largest gains on correlation geometry: PC1 variance share (+0.17) and effective rank (+0.09). These descriptors capture factor concentration and dimensionality of the return distribution, precisely the cross-sectional structure that distinguishes market regimes. The volatility gains (RV21 +0.08, RV5 +0.09) are smaller but consistent. Critically, both models remain weak on direction descriptors (market return $R^2 < 0.1$), confirming that neither objective learns to predict returns.

*formly superior*: its clearest gain is on slow diversification recovery, with little benefit on stress onsets or directional moves. As a label-free stress indicator, embedding displacement matches the strongest standard risk measures, though it is largely redundant with a full risk bundle. Without text supervision, latent regimes align with external news semantics beyond random, seasonality, and volatility-only baselines.

We view this work as a foundation for further explorations in applying Joint-Embedding Predictive Architectures to regime-level dynamics within financial markets.

### 5.1. Future work

**Forecasting/counterfactual prediction.** In V-JEPA 2 (Assran et al., 2025), the authors introduce an action-conditioned training stage that utilizes a frozen encoder to predict future world states from robot states and proposed actions. Exploring the possibility of such counterfactual prediction using latent representations of the equities and high-level labels—such as news headlines or sentiment signals derived from prediction markets—is a natural extension of this work. In this setting, alternative policy decisions or information shocks can be treated as interventions on the latent action space, enabling "what-if" analyses of market dynamics without requiring explicit reconstruction in input space.

**Scaling.** Computational constraints limited our exploration of larger or more advanced tokenizers, deeper encoders, more complex stride curricula, ablations across features, higher-fidelity or denser data, and longer context windows. Expanding on any one of these axes would be an interesting direction of work. For our evaluations, we primarily utilize ridge/logistic regression, nonlinear decoders might recover more signal. Considering our data universe, we evaluate only U.S. equities from 2018–2025, where top-ADV selection excludes less liquid assets. We leave exploration of generalization to other markets or eras for future work.

**Further interpretability of market geometry.** Although volatility is a dominant driver of correlation geometry, there are economically meaningful periods in which correlation structure shifts without large changes in volatility (e.g., sectoral rotations or factor crowding). A natural extension is to train or evaluate representations specifically on these episodes, or to design objectives that emphasize changes in correlation shape conditional on volatility.

## Impact Statement

This work studies the feasibility of learning compact, day-level representations of U.S. equity markets from cross-sectional price and volume data using a self-supervised approach. The method is descriptive rather than predictive:

it produces embeddings that summarize persistent second-moment structure (volatility and correlation geometry) in historical market data, and is intended for analysis, monitoring, and retrospective characterization rather than decision-making or forecasting.

Potential positive impacts include: (i) providing a compact, day-level representation of market-wide second-moment structure that can serve as a foundation for empirical study, without requiring direct manipulation of high-dimensional covariance objects; (ii) offering a simple summary of historical volatility and correlation patterns that may be useful for descriptive monitoring or regime comparison in retrospective analyses; and (iii) producing latent representations that can be used as inputs to other analytical or modeling systems, where a low-dimensional market state description is required.

At the same time, misuse or over-interpretation of such representations poses risks. First, we make no claims that the embeddings are suitable for trading, price prediction, return forecasting, market timing, or alpha generation; the method is not evaluated in any predictive or trading setting, and treating the embeddings as decision signals could be misleading. Second, because the embeddings emphasize slow-moving structure, they are not designed to detect or anticipate abrupt market stress, and reliance on them for early-warning or real-time risk management could create false confidence. Third, as with other market-level summaries, widespread deployment could reinforce crowding or informational asymmetries if used without appropriate governance.

All experiments rely exclusively on publicly observable market data sourced via a commercial vendor (Massive.com) and aggregated news metadata (GDELT), and do not involve personal or sensitive information. We encourage responsible use through careful out-of-sample evaluation, clear documentation of intended use and limitations, and deployment only in settings where model governance and oversight are appropriate to the stakes involved.

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

## A. Data and Features

### A.1. Universe and Splits

We use daily OHLCV data from Massive.com for U.S. common equities listed on NYSE, NASDAQ, and NYSE American. The universe consists of the top 512 equities by average dollar volume, computed on training data only to prevent lookahead bias. Raw data spans October 2017 through October 2025 (2,010 trading days); feature warmup (126-day lookback for momentum features) and clip burn-in ($L{=}21$) consume the initial $\sim$150 days, yielding an effective evaluation period of May 2018 onward (1,864 days).

**Train/validation/test splits.** We use a strict temporal split with no overlap: training covers 2018–2022, validation covers 2023, and test covers 2024–2025. All feature normalization statistics (z-score means and standard deviations) are computed on the training split only. Assets are selected once at the start of training and held fixed; we do not rebalance the universe over time.

### A.2. Feature Computation

Table 7 defines the $F{=}28$ per-equity features. All returns are computed from split-adjusted close prices (close-to-close basis); dividend adjustments are not applied. Rolling windows require a minimum number of valid observations as noted; assets with insufficient history on a given day receive NaN values, which are replaced with zeros during batching while a validity mask excludes them from attention.

**Notation.** $O_t, H_t, L_t, C_t, V_t$ denote open, high, low, close, and volume on day $t$. The constant $\varepsilon = 10^{-8}$ prevents division by zero. EWMA decay factors are $\alpha_h = 1 - \exp(-\ln 2/h)$ for half-life $h$.

## B. Model Architecture

### B.1. Tokenizer Architecture

The tokenizer maps the variable-size daily cross-section $\mathbf{X}_t \in \mathbb{R}^{N_t \times F}$ to a fixed set of $K{=}24$ factor tokens $\mathbf{T}_t \in \mathbb{R}^{K \times d}$ with $d{=}128$. The architecture follows (Lee et al., 2019):

1. **Row-wise MLP**: $\psi : \mathbb{R}^F \to \mathbb{R}^d$ embeds each equity independently:

$$\mathbf{e}_{t,i} = \psi(\mathbf{x}_{t,i}) \in \mathbb{R}^d, \quad i = 1, \ldots, N_{\max}.$$

2. **ISAB stack**: $B_{\text{ISAB}}{=}2$ inducing-point set attention blocks with $m{=}32$ inducing points enable cross-equity interaction:

$$\mathbf{Z}_t = \text{ISAB}_m^{(B_{\text{ISAB}})}(\mathbf{E}_t) \in \mathbb{R}^{N_{\max} \times d},$$

scaling as $O(B_{\text{ISAB}} \cdot N_{\max}m)$ rather than $O(N_{\max}^2)$.

3. **PMA**: Pooling by multihead attention with $K{=}24$ learned seed vectors $\mathbf{S} \in \mathbb{R}^{K \times d}$:

$$\mathbf{U}_t = \text{PMA}_K(\mathbf{Z}_t) = \text{MAB}(\mathbf{S}, \mathbf{Z}_t) \in \mathbb{R}^{K \times d}.$$

Each seed can attend to a different subset of equities, yielding learned "factor" summaries.

4. **SAB**: A final self-attention block models dependencies among the $K$ tokens:

$$\mathbf{T}_t = \text{SAB}(\mathbf{U}_t) \in \mathbb{R}^{K \times d}.$$

All attention blocks use 4 heads, LayerNorm, and dropout rate 0.1.

**Handling missing assets.** In implementation, we pad to $N_{\max}{=}512$ and apply a validity mask $\mathbf{m}_t \in \{0, 1\}^{N_{\max}}$. Invalid keys receive $-\infty$ logits before softmax, ensuring attention is computed only over live assets. Padded rows may still be updated as queries in intermediate blocks, but because they are excluded as keys/values, they do not influence the pooled tokens.

*Table 7.* **Equity feature vector** ($F$=28). Each feature is computed per equity per trading day from adjusted daily OHLCV bars. All features are z-scored using statistics from the *training split only*. Features with insufficient history receive NaN (replaced with zero under a validity mask).

| Feature | Definition | Window |
|---|---|---|
| **Returns & horizon structure** | | |
| log_ret_1 | $\log(C_t/C_{t-1})$ | 1d |
| log_ret_5 | $\log(C_t/C_{t-5})$ | 5d |
| log_ret_21 | $\log(C_t/C_{t-21})$ | 21d |
| log_ret_63 | $\log(C_t/C_{t-63})$ | 63d |
| log_ret_126 | $\log(C_t/C_{t-126})$ | 126d |
| overnight_gap | $\log(O_t/C_{t-1})$ | overnight |
| intraday_ret | $\log(C_t/O_t)$ | intraday |
| mom_6_1 | log_ret_126 $-$ log_ret_21 | momentum |
| **Volatility & regime** | | |
| rvol_10 | $\mathrm{Std}($log_ret_1$)$ | 10d |
| rvol_21 | $\mathrm{Std}($log_ret_1$)$ | 21d |
| rvol_63 | $\mathrm{Std}($log_ret_1$)$ | 63d |
| rvol_ratio | rvol_10$/($rvol_63$+\varepsilon)$ | regime |
| ewma_vol_hl10 | $\sqrt{\mathrm{EWMA}(\texttt{log\_ret\_1}^2;\alpha_{10})}$ | HL=10 |
| ewma_vol_hl20 | $\sqrt{\mathrm{EWMA}(\texttt{log\_ret\_1}^2;\alpha_{20})}$ | HL=20 |
| **OHLC candle geometry** | | |
| hl_log_range | $\log(H_t/L_t)$ | intraday |
| body_to_range | $|\log(C_t/O_t)|/(\log(H_t/L_t)+\varepsilon)$ | normalized |
| upper_shadow | $\log\big(H_t/\max(O_t,C_t)\big)$ | wick |
| lower_shadow | $\log\big(\min(O_t,C_t)/L_t\big)$ | wick |
| **Volume & liquidity** | | |
| log_volume | $\log(1+V_t)$ | |
| log_dollar_volume | $\log(1+C_tV_t)$ | dollar |
| rel_dvol_21 | $(C_tV_t)/(\mathrm{Mean}_{21}(CV)+\varepsilon)$ | relative |
| dvol_z_21 | $(CV-\mathrm{Mean}_{21}(CV))/(\mathrm{Std}_{21}(CV)+\varepsilon)$ | z-score |
| amihud_illiq_1 | $|$log_ret_1$|/(C_tV_t+\varepsilon)$ | Amihud |
| amihud_illiq_21 | $\mathrm{Mean}_{21}($amihud_illiq_1$)$ | 21d avg |
| **Market-relative** | | |
| mkt_log_ret_1 | mean of log_ret_1 across active assets | XS mean |
| ret_vs_mkt_1 | log_ret_1 $-$ mkt_log_ret_1 | residual |
| mkt_rvol_21 | mean of rvol_21 across active assets | XS mean |
| rel_vol | rvol_21$/($mkt_rvol_21$+\varepsilon)$ | vol ratio |

## B.2. JEPA Architecture

The temporal encoder operates on $L$=21-day clips of factor tokens. Both the context encoder and predictor are 6-layer transformers with 4 attention heads, MLP ratio 4.0, and drop-path rate 0.1. The predictor uses a narrower hidden dimension (64 vs. 128). Positional encodings use Fourier features with logarithmically spaced frequencies, encoding calendar distance rather than sequence index.

# C. Training Details

### C.1. Hyperparameters

Table 8 summarizes the key hyperparameters for both the tokenizer and temporal encoder.

**Learning rate schedule.** We use a warmup-stable-decay (WSD) schedule following V-JEPA 2: linear warmup ($10^{-6} \to 5\times10^{-4}$) over 10 epochs, constant for 170 epochs, then linear decay to $10^{-5}$ over 20 epochs. Optimizer is AdamW with $(\beta_1, \beta_2) = (0.9, 0.999)$.

*Table 8.* **Model and training hyperparameters.**

| Component | Parameter | Value |
|---|---|---|
| Tokenizer | Input dimension $F$ | 28 |
| | Hidden dimension $d$ | 128 |
| | Factor tokens $K$ | 24 |
| | Inducing points $m$ | 32 |
| | ISAB layers | 2 |
| JEPA encoder | Embedding dimension | 128 |
| | Encoder depth | 6 |
| | Predictor depth | 6 |
| | Predictor dimension | 64 |
| | Attention heads | 4 |
| | MLP ratio | 4.0 |
| Data | Clip length $L$ | 21 |
| | Max assets $N_{\max}$ | 512 |
| | Batch size | 128 |
| | Samples per epoch | 8192 |
| Optimization | Total epochs | 200 |
| | Peak learning rate | $5 \times 10^{-4}$ |
| | Warmup / final LR | $10^{-6}$ / $10^{-5}$ |
| | Warmup / anneal epochs | 10 / 20 |
| | Weight decay | 0.04 |
| | Gradient clipping | 1.0 |
| EMA teacher | Momentum (start $\rightarrow$ end) | $0.996 \rightarrow 0.9999$ |
| | Schedule | Linear ramp |

## C.2. Stride Curriculum and Masking

Table 9 details the stride curriculum, which progressively introduces longer temporal horizons during training, and the masking strategy.

## C.3. Implementation Details

**Software environment.** All experiments use PyTorch 2.0+ with CUDA 12. Key dependencies: `torch`$\geq$2.0, `numpy`$\geq$1.24, `zarr`$\geq$2.16 (data storage), `wandb`$\geq$0.15 (experiment tracking). Training uses `bfloat16` mixed precision via PyTorch AMP.

**Data pipeline.** Input features are stored in Zarr format as $\mathcal{X} \in \mathbb{R}^{T \times N_{\max} \times F}$ ($T$=2,010 days, $N_{\max}$=512 assets, $F$=28 features). Normalization is two-stage: (i) per-day cross-sectional robust scaling (median, MAD) to handle outliers within each day's cross-section, this uses same-day data only; (ii) temporal z-scoring using means and standard deviations computed on the *training split only*. The entire dataset is cached in memory during training, eliminating I/O bottlenecks.

**Position embeddings.** Temporal positions encode *calendar distance* $\Delta(\ell) = (L - 1 - \ell) \times s$ (trading days before clip endpoint) via Fourier features with log-spaced frequencies ($\omega_{\max}$=10), followed by a learned projection. This enables smooth interpolation across stride values. Slot embeddings (distinguishing $K$ tokens per time step) are learned.

**Initialization.** Following V-JEPA 2, output projections in attention and MLP blocks are rescaled by $1/\sqrt{2\ell}$ (layer $\ell$). Other weights use truncated normal ($\sigma$=0.02).

## C.4. Compute

**Hardware.** All experiments were conducted on a single NVIDIA H100 GPU with 80GB memory. No multi-GPU parallelism was required.

*Table 9.* **Stride curriculum and masking strategy.**

| Phase | Setting | Value |
|---|---|---|
| *Stride curriculum* | | |
| Early (epochs 1–30) | Strides | $\{1\}$ |
| Mid (epochs 31–80) | Strides | $\{1, 3, 7\}$ |
| | Sampling weights | $(0.5, 0.3, 0.2)$ |
| Late (epochs 81–200) | Strides | $\{1, 3, 7, 21\}$ |
| | Sampling weights | $(0.3, 0.3, 0.2, 0.2)$ |
| *Masking* | | |
| | Minimum visible ratio | 0.3 |
| | Causal masking probability | 0.3 |
| | Mask structure | Spatiotemporal blocks |
| | Mask token | Learnable |

**Main model training.** The jointly-trained JEPA model (tokenizer + encoder) was trained for 200 epochs with batch size 128, completing in approximately 12 hours wall-clock time. Peak memory usage was $\sim$20GB, well within the H100's capacity. Each epoch consists of 64 gradient steps (8192 samples / 128 batch size).

**Evaluation.** Latent extraction for all $T_{\text{feat}}$=1,884 days requires approximately 10 minutes on CPU. The kNN evaluation, geometry fidelity analysis, and event detection together require approximately 5 minutes per model.

### C.5. Masking Strategy Ablation Details

The masking strategy ablation in Section 4.2 compares the main model's structured masking against random i.i.d. masking. Both models share identical architecture and training configuration; they differ only in how masked positions are selected.

**Structured masking (main model).** Masks are contiguous in time but random over factor slots, with a mixture of: (i) time-span $\times$ random-slot masks, (ii) full-day masks (all slots on selected days), and (iii) full-slot masks (selected slots across all time). This encourages the model to predict from temporally distant context, emphasizing persistent cross-sectional patterns.

**Random masking (ablation).** Mask positions are sampled uniformly across the $L \times K$ grid without temporal structure. Both strategies maintain a minimum visible ratio of 0.3.

### C.6. Reproducibility

**Random seeds.** All experiments use a fixed seed (42) for PyTorch, NumPy, and Python's random module. The CUDA deterministic flag is enabled where possible, though some operations (e.g., atomicAdd in attention) may introduce minor non-determinism.

**Data splits.** The dataset is split chronologically with no overlap: training covers 2018–2022 (1,309 trading days), validation covers 2023 (250 days), and test covers 2024–2025 (451 days through October 2025). Splits are defined by index boundaries in the manifest file, ensuring exact reproducibility.

**Checkpointing.** Model checkpoints are saved every 5 epochs, with the best checkpoint (lowest validation loss) retained separately. Checkpoints include full optimizer state for training resumption.

## D. Additional Experiments

### D.1. No-Volatility-Channels Retrain

We retrain the pipeline (tokenizer + JEPA encoder + predictor) end-to-end with all eight explicit volatility input features removed (`rvol_10`, `rvol_21`, `rvol_63`, `rvol_ratio`, `ewma_vol_hl10`, `ewma_vol_hl20`, `mkt_rvol_21`, `rel_vol`), leaving $F = 20$ inputs. All other features, architecture, training hyperparameters, and splits are identical to the

main configuration. This tests whether the embedding's correlation-geometry signal is merely a passthrough of the explicit volatility channels.

Table 10 reports geometry, recovery, and news metrics for both models.

*Table 10.* **No-volatility-channels retrain.** All eight explicit volatility input features removed; tokenizer + JEPA retrained end-to-end. Recovery AUC gain reported on `recovery_eff_rank_UP` (incremental over the geometry-scalar baseline). Geometry $R^2$ averaged across `corr_mean`, `pc1_share`, `eff_rank`.

| Metric | JEPA (main) | JEPA (no-vol) |
|---|---|---|
| Mantel $\rho$ (gap $\geq$ 63d) | 0.348 | 0.321 |
| Mean geometry $R^2$ | 0.408 | 0.378 |
| Recovery AUC gain | +0.243 | +0.138 |
| News $\Delta(63)$ (train+val anchor) | 0.0053 | 0.0080 |

Removing the explicit volatility channels weakens the geometry, recovery, and news signals but does not eliminate them, ruling out a simple feature-passthrough explanation. The recovery AUC gain roughly halves but stays positive, and the $W=63$ news alignment is comparable to the main model.

### D.2. Regime Classification with HMM

To assess whether the learned representations capture market regime information, we fit a Gaussian Hidden Markov Model on market observables (equal-weighted returns, realized volatility, and cross-sectional dispersion) and evaluate whether a linear probe can recover regime labels from the market-state embedding alone.

The HMM identifies four regimes via validation likelihood: low-volatility (42% of days), mid-volatility (40%), high-volatility (15%), and crisis (3.5%). We obtain hard labels using the Viterbi algorithm and train a logistic regression probe on the market-state embedding $\mathbf{z}_t \in \mathbb{R}^{128}$.

*Table 11.* **Regime classification on held-out test set** (451 days, 2024–2025). 95% CI from 1000 bootstrap samples.

| | Accuracy | F1 (macro) | AUROC |
|---|---|---|---|
| JEPA | 0.633 [0.59, 0.68] | 0.386 [0.35, 0.43] | 0.843 [0.80, 0.88] |
| Observables | 0.895 [0.87, 0.92] | 0.641 [0.55, 0.73] | 0.966 [0.93, 0.99] |
| Random | 0.250 | — | 0.500 |

The latent probe significantly outperforms chance ($p < 10^{-4}$, permutation test), confirming that the representations encode regime-relevant structure. The observable baseline achieves higher accuracy, which is expected given that regimes are defined on these same observables.

We additionally test whether latent displacements $\|\mathbf{z}_t - \mathbf{z}_{t-1}\|_2$ can detect regime transitions. At the top 5% of displacement magnitudes, 39% correspond to actual regime changes versus a 6.4% base rate (6.1$\times$ lift). A lag sweep finds optimal alignment at lag 0, indicating that latent dynamics are synchronous with—rather than predictive of—regime shifts.

These results suggest the encoder learns a representation that tracks market conditions, though the signal is contemporaneous rather than anticipatory.

### D.3. Event Detection: Extended Results

Further investigation is outlined in Table 12.

### D.4. Displacement vs. Standard Risk Measures

We compare embedding displacement $\delta_t = \|\mathbf{z}_t - \mathbf{z}_{t-1}\|_2$ against eight standard risk signals on the held-out test split (2024–2025), using the same Crash+VIX and All-Systematic stress labels as Section 3.4.

**Benchmark signals.** |SPX return| (absolute daily index return), 5d rolling SPX volatility, 63d SPX drawdown (decline from trailing peak), VIX level, |$\Delta$VIX| (absolute daily VIX change), HY spread level and daily change (ICE BofA US

*Table 12.* Event detection via latent displacement. Enrichment reports appearance in top 1% of displacements; displacement reports weekday-adjusted latent jump magnitude.

| Category | $N$ | Enrichment (Top 1%) | | | | Displacement | | |
| | | Hits | Rate | OR | $p$ | Effect | $d$ | $p$ |
|---|---|---|---|---|---|---|---|---|
| FOMC | 66 | 1 | 1.5% | 1.5 | .50 | 0.12 | 0.06 | .63 |
| CPI | 89 | 1 | 1.1% | 1.1 | .61 | 0.06 | 0.03 | .79 |
| Crash Days | 30 | 5 | 16.7% | 26.0 | <.001 | 1.78 | 0.92 | <.001 |
| VIX Spikes | 46 | 8 | 17.4% | 34.6 | <.001 | 2.80 | 1.47 | <.001 |
| Market Stress | 59 | 9 | 15.3% | 32.3 | <.001 | 2.26 | 1.18 | <.001 |
| All Systematic | 203 | 9 | 4.4% | 7.7 | <.001 | 0.73 | 0.37 | <.001 |

High-Yield OAS), and a cross-sectional liquidity proxy (median log dollar volume across the active universe).

**Standalone enrichment.** Table 13 reports the top-1% enrichment lift of each individual signal on Crash+VIX days. JEPA displacement ties the strongest individual signal (|SPX return|) at $14.96\times$.

*Table 13.* **Top-1% enrichment lift on Crash+VIX days** (test 2024–2025). Higher is stronger.

| Signal | Lift ($\times$) |
|---|---|
| JEPA displacement $\delta_t$ | 14.96 |
| \|SPX return\| | 14.96 |
| VIX level | 13.30 |
| 63d SPX drawdown | 11.63 |
| 5d rolling SPX vol | 9.97 |
| \|$\Delta$VIX\| | 8.31 |
| HY spread level | 6.65 |
| HY spread change | 5.82 |
| Liquidity (med. log $vol) | 4.99 |

**Incremental information beyond the full risk bundle.** Table 14 reports logistic regressions on the same Crash+VIX and All-Systematic labels comparing (i) the full risk bundle alone and (ii) the bundle augmented with JEPA displacement. We report ROC-AUC and average precision with 95% paired-bootstrap CIs on the held-out test split.

*Table 14.* **Incremental value of JEPA displacement beyond the standard risk bundle** (test 2024–2025). $\Delta$ is the paired difference (bundle $+ \delta_t$) $-$ (bundle alone) with 95% bootstrap CI. CIs covering zero indicate the gain is not statistically distinguishable from zero.

| Label | Bundle (AUC / AP) | Bundle $+\delta_t$ | $\Delta$ AUC [95% CI] | $\Delta$ AP [95% CI] |
|---|---|---|---|---|
| Crash + VIX | 0.985 / 0.586 | 0.983 / 0.595 | $-0.002$ [$-.006, .003$] | $+0.009$ [$-.027, .045$] |
| All Systematic | 0.716 / 0.298 | 0.715 / 0.303 | $-0.001$ [$-.011, .009$] | $+0.005$ [$-.014, .023$] |

## D.5. Cross-Modal Baseline Comparison

Table 15 reports the full $\Delta(W)$ curves for the JEPA market-state embedding and a calendar seasonality baseline (day-of-week + month indicators). The JEPA representation achieves statistically significant news alignment at exclusion windows $W \geq 126$ days on the train+validation period, with $W = 63$ marginal (CI boundary near zero); effect sizes increase monotonically from $\Delta(126) = 0.0060$ to $\Delta(252) = 0.0069$. This replicates on the held-out test period (2024–2025), where significance emerges earlier ($W \geq 21$) and the peak effect at $W = 126$ reaches $\Delta = 0.0162$, nearly three times the train+validation estimate. The seasonality baseline never achieves significance: its confidence intervals include zero at all windows, confirming that the cross-modal alignment captured by JEPA cannot be explained by simple calendar effects.

## D.6. Extended Geometry Transition Results

Table 16 reports the full level-decoding diagnostic confirming that the market-state embedding encodes contemporaneous geometry.

*Table 15.* **News retrieval $\Delta(W)$: market-state embedding vs. calendar seasonality baseline.** 95% bootstrap CI. Train+Val: 2018–2023, Test: 2024–2025.

| Method | Period | Exclusion Window $W$ (days) | | | | | |
| | | 0 | 5 | 21 | 63 | 126 | 252 |
|---|---|---|---|---|---|---|---|
| **JEPA** | Train+Val | 0.0021 | 0.0043 | 0.0049 | $0.0053^{\dagger}$ | 0.0060* | 0.0069* |
| | (95% CI) | [-.003, .008] | [-.002, .011] | [-.000, .010] | [-.000, .011] | [.001, .013] | [.002, .013] |
| | Test | 0.0047 | 0.0061 | 0.0057* | 0.0077* | 0.0162* | 0.0077* |
| | (95% CI) | [-.001, .008] | [-.001, .011] | [.001, .010] | [.003, .016] | [.012, .026] | [.002, .016] |
| Seasonality | Train+Val | -0.0004 | 0.0003 | 0.0002 | 0.0029 | 0.0032 | 0.0057 |
| | (95% CI) | [-.005, .004] | [-.005, .008] | [-.005, .008] | [-.003, .012] | [-.003, .011] | [-.001, .015] |

\* Significant at 95% (CI excludes zero); $^{\dagger}$ marginal (CI boundary near zero).

*Table 16.* **Level decoding diagnostic.** Ridge regression from $\mathbf{z}_t$ to contemporaneous geometry metrics on held-out test data ($n_{\text{test}} = 365$). All Spearman correlations significant at $p < 10^{-20}$.

| Target | $R^2$ | Spearman $\rho$ |
|---|---|---|
| corr_mean | 0.415 | 0.499 |
| pc1_share | 0.447 | 0.615 |
| eff_rank | 0.361 | 0.577 |

**Stronger baselines for transition prediction.** Table 17 reports JEPA's incremental value above an 18-feature handcrafted baseline (six descriptors – logRV21, dispersion, breadth, left tail, pc1_share, eff_rank – each augmented with daily/weekly/monthly persistence summaries) on the four transition tasks. The same logistic-regression protocol as Section 3.3.2 is used, with a 21-day purge buffer. HAR (Corsi, 2009) and a Gaussian HMM on market observables are domain references; MAE (matched tokenizer, encoder, data, and split) is the capacity-matched learned alternative.

*Table 17.* **Transition prediction under domain baselines.** ROC-AUC / average precision on held-out test data. "Base" is the 18-feature handcrafted persistence baseline; "+ JEPA" and "+ MAE" add the corresponding learned market-state embedding. $\Delta$ values are absolute (AUC and AP). Positive numbers favor adding the learned representation.

| Task | Base | + JEPA | + MAE | $\Delta$ JEPA | $\Delta$ MAE |
|---|---|---|---|---|---|
| recovery_eff_rank_UP | 0.690 / 0.139 | 0.800 / 0.324 | 0.723 / 0.200 | +0.110/ + 0.185 | +0.033/ + 0.061 |
| stress_corr_mean_UP | 0.549 / 0.085 | 0.671 / 0.093 | 0.541 / 0.067 | +0.122/ + 0.008 | −0.008/ − 0.018 |
| recovery_corr_mean_DOWN | 0.969 / 0.818 | 0.960 / 0.683 | 0.973 / 0.887 | −0.008/ − 0.136 | +0.004/ + 0.069 |
| recovery_pc1_share_DOWN | 0.972 / 0.837 | 0.754 / 0.360 | 0.937 / 0.770 | −0.218/ − 0.476 | −0.035/ − 0.067 |

JEPA's clearest additive value survives on recovery_eff_rank_UP, with a smaller AUC gain on stress_corr_mean_UP. The high-baseline recovery_corr_mean_DOWN and recovery_pc1_share_DOWN tasks are saturated by the persistence features and adding JEPA hurts the latter. Latent prediction adds information about slow diversification dynamics that handcrafted persistence summaries do not capture, and offers no advantage when the descriptor itself is highly persistent.

### D.7. JEPA vs MAE Ablation: Extended

We provide extended results for the JEPA vs MAE comparison summarized in Table 5.

**Experimental controls.** Both models use identical configurations except for the training objective:

- **Tokenizer**: Shared pretrained tokenizer (K=24 tokens, d=128, 2 ISAB layers). Frozen during encoder training.

- **Encoder**: 6-layer transformer, 4 heads, MLP ratio 4.0, dim=128.

- **Data**: Same temporal clips (L=21 days), same train/test split (2018–2022 train, 2024–2025 test).

- **Optimization**: AdamW, peak LR $5 \times 10^{-4}$, 200 epochs, cosine schedule.

- **Masking**: Structured spatiotemporal masks with 30% minimum visible ratio.

- **Encoder masking**: The MAE encoder is visible-only, following the standard protocol (He et al., 2021); masked positions are reintroduced as learned mask tokens at the decoder, not the encoder.

The MAE baseline uses a 4-layer decoder (dim=128) that reconstructs masked token features via MSE loss. JEPA uses a 6-layer predictor (dim=64) that predicts EMA target embeddings via $\ell_1$ loss.

**Full kNN retrieval results.** Table 18 reports $R^2$ for both methods across neighborhood sizes $k \in \{5, 10, 20\}$. We use the $\pm 21$d exclusion window throughout to prevent temporal leakage.

*Table 18.* **kNN retrieval $R^2$ across neighborhood sizes** ($\pm 21$d exclusion). Bold indicates best per descriptor.

| Descriptor | JEPA | | | MAE | | |
| | $k=5$ | $k=10$ | $k=20$ | $k=5$ | $k=10$ | $k=20$ |
|---|---|---|---|---|---|---|
| RV21 | 0.52 | 0.57 | **0.62** | 0.43 | 0.48 | 0.54 |
| RV5 | 0.27 | 0.37 | **0.41** | 0.14 | 0.25 | 0.32 |
| PC1 share | 0.03 | 0.19 | **0.29** | $-0.19$ | $-0.04$ | 0.12 |
| Eff. rank | 0.01 | 0.15 | **0.24** | $-0.13$ | 0.02 | 0.15 |
| Dispersion | 0.14 | 0.16 | **0.21** | 0.07 | 0.13 | 0.15 |
| Left tail | 0.16 | 0.24 | **0.27** | 0.25 | 0.24 | 0.23 |
| Mkt. return | $-0.08$ | 0.04 | 0.07 | 0.01 | 0.03 | 0.05 |
| Breadth | $-0.13$ | $-0.01$ | 0.02 | $-0.08$ | $-0.02$ | 0.02 |

Several patterns emerge. First, $R^2$ increases monotonically with $k$ for both methods, reflecting the variance reduction from averaging over more neighbors. Second, JEPA's advantage over MAE is largest for correlation-geometry descriptors (PC1 share, effective rank) and smallest for tail risk (left tail). Third, MAE exhibits negative $R^2$ at small $k$ for several descriptors—its neighbors are worse than predicting the mean—while JEPA remains positive or near-zero.

**Comparison to baselines.** Table 19 compares both methods against three baselines: (i) *lag-1 persistence*, which predicts today's descriptor equals yesterday's; (ii) *time-neighbor*, which retrieves the $k$ temporally nearest days; and (iii) *random*, which uses random 128-dimensional embeddings.

*Table 19.* **Baseline comparison** ($k=20$, $\pm 21$d exclusion). "Beats" indicates methods where the row method achieves higher $R^2$.

| Descriptor | JEPA | MAE | Lag-1 | Time | Random |
|---|---|---|---|---|---|
| RV21 | **0.62** | 0.54 | 0.97 | $-1.55$ | $-0.29$ |
| RV5 | **0.41** | 0.32 | 0.67 | $-1.07$ | $-0.19$ |
| PC1 share | **0.29** | 0.12 | 0.96 | $-0.49$ | $-0.51$ |
| Eff. rank | **0.24** | 0.15 | 0.95 | $-0.83$ | $-0.53$ |
| Dispersion | **0.21** | 0.15 | $-0.58$ | $-0.02$ | $-0.20$ |
| Left tail | **0.27** | 0.23 | $-0.78$ | $-0.04$ | $-0.12$ |
| Mkt. return | 0.07 | 0.05 | $-1.14$ | $-0.00$ | $-0.05$ |
| Breadth | 0.02 | 0.02 | $-1.11$ | $-0.00$ | $-0.06$ |

Both JEPA and MAE beat the time-neighbor and random baselines on all structure descriptors. The lag-1 baseline achieves high $R^2$ for volatility (RV21, RV5) and correlation structure (PC1 share, effective rank) due to their persistence, but fails on dispersion and left tail. Neither learned method approaches lag-1 performance on highly persistent descriptors—this is expected, as kNN retrieval spans years while lag-1 exploits day-to-day autocorrelation.

The time-neighbor baseline yields negative $R^2$ on most descriptors (median gap >900 days with the exclusion window), confirming that temporal proximity alone does not explain the learned neighborhoods. JEPA retrieves days with similar risk geometry regardless of calendar distance.

**Time gap statistics.** With the $\pm 21$d exclusion, the median temporal gap between query days and their $k=20$ neighbors is 924 days for JEPA and 885 days for MAE. Both methods retrieve neighbors spanning the full evaluation period (2018–2025),

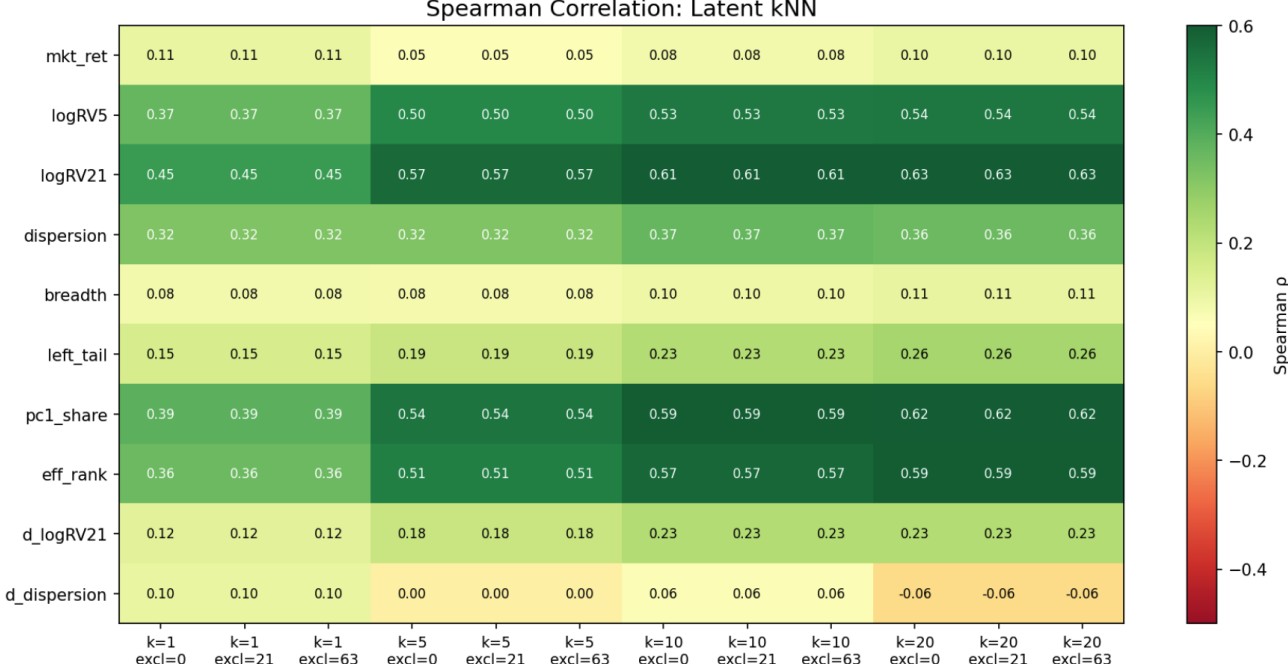

*Figure 5.* **Extended kNN retrieval evaluation.** Spearman rank correlations between query-day descriptors and neighbor-based estimates across neighborhood sizes and temporal exclusion windows. Rows correspond to market descriptors; columns vary $k$ and exclusion width.

with 0% of neighbors falling within 63 trading days of the query. This rules out temporal leakage as an explanation for the retrieval performance.

### D.8. kNN Retrieval: Sensitivity Analysis

Figure 5 reports an extended version of the kNN retrieval evaluation from the main text, varying both the neighborhood size ($k \in \{1, 5, 10, 20\}$) and the temporal exclusion window ($\pm\{0, 21, 63\}$ trading days). Each cell shows the Spearman rank correlation between a query-day descriptor and the corresponding descriptor averaged over its latent nearest neighbors.

Across all settings, descriptors associated with cross-sectional risk structure—such as realized volatility (RV5, RV21), principal component variance share, and effective rank—exhibit consistently strong correlations that increase with neighborhood size. In contrast, directional quantities such as the contemporaneous market return remain weakly correlated across all configurations, indicating that the latent space does not primarily encode short-horizon directional information.

Importantly, correlations for volatility and correlation-geometry measures remain stable under increasingly aggressive temporal exclusion, suggesting that the learned latent neighborhoods reflect persistent market states rather than local temporal proximity. First-difference descriptors (e.g., $\Delta$RV21 and $\Delta$dispersion) show systematically lower correlations, consistent with the encoder emphasizing level effects over high-frequency changes.

Overall, this sensitivity analysis supports the interpretation that the market-state embedding organizes days according to enduring cross-sectional risk structure, and that the kNN results reported in the main text are not driven by a particular choice of $k$ or exclusion window.

## E. GDELT News Data

We use the GDELT Global Knowledge Graph (GKG) v2 via Google BigQuery (`gdelt-bq.gdeltv2.gkg`), extracting articles from 2017–2025 that mention U.S. locations. Timestamps are converted to Eastern Time and shifted by $-16$ hours so that the "news day $t$" boundary falls at approximately 16:00 ET (market close): news published from 16:00 ET on day $t-1$ through 15:59 ET on day $t$ is assigned to market date $t$. We exclude generic theme prefixes (`CRISISLEX_*`, `GENERAL_*`, `TAX_*`, `WB_*`) and individual themes (`LEADER`, `MEDIA_MSM`, `AFFECT`).

For each (date, theme) pair, we compute document counts and volume shares. The dataset contains 2.58M rows covering 1,263 themes across 2,010 trading days. Themes are aggregated to families by first underscore-delimited prefix (e.g., ECON_BANKRUPTCY → ECON), yielding 168 families. We retain the eight largest (Table 20). Daily family vectors use $\log(1 + \texttt{count})$ weighting with L2 normalization.

*Table 20.* **GDELT theme families.** Count = unique themes per family.

| Family | Count | Examples |
|---|---|---|
| ECON | 491 | ECON_BANKRUPTCY, ECON_INFLATION |
| NATURAL | 186 | NATURAL_DISASTER_WILDFIRE, NATURAL_DISASTER_FLOOD |
| MANMADE | 118 | MANMADE_DISASTER_CHEMICAL, MANMADE_DISASTER_NUCLEAR |
| ENV | 22 | ENV_CLIMATECHANGE, ENV_SOLAR |
| UNREST | 18 | UNREST_PROTEST, UNREST_RIOT |
| GOV | 11 | GOV_REFORM, GOV_ELECTION |
| EMERG | 8 | EMERG_PANDEMIC, EMERG_DISEASE |
| MED | 7 | MED_HEALTH, MED_DISEASE |

# F. Evaluation Metrics and Financial Glossary

We define the shorthand used in the evaluation suite (Table 1) and the financial quantities used as descriptors and benchmark signals, for readers without a finance background.

## F.1. Evaluation metrics (Table 1)

**Ridge $R^2$:** fit of an L2-regularized linear probe from $\mathbf{z}_t$ to a target (1 perfect, 0 predicts the mean, negative worse).

**Spearman $\rho$:** rank correlation; robust to outliers and nonlinearity.

**$\pm 21$d exclusion:** kNN neighbors within 21 trading days of the query are dropped to avoid temporal-adjacency matches.

**Mantel $\rho$:** Spearman correlation between the upper triangles of the latent- and geometry-distance matrices.

**gap $\geq 63$d:** the Mantel test uses only day-pairs at least 63 trading days apart.

**% improvement:** reduction in geometry distance for latent neighbors versus random days.

**$\Delta$AUC, $\Delta$PR-AUC:** gain in ROC-AUC and precision–recall AUC (average precision) from adding $\mathbf{z}_t$ to a baseline; PR-AUC suits the rare-event imbalance.

**Enrichment OR:** odds ratio of a labeled event among top-percentile displacement days versus the rest (1 = none; Fisher's exact test).

**$\Delta(W)$:** news similarity of latent neighbors minus time-shuffled neighbors at window $W$; positive means shared news beyond temporal proximity.

## F.2. Financial terms

**Realized volatility (RV):** standard deviation of daily log returns over a trailing window (RV21: 21d, RV5: 5d); higher is more turbulent.

**Mean correlation (`corr_mean`):** average pairwise return correlation; high means stocks move together (poor diversification).

**PC1 share (`pc1_share`):** variance fraction explained by the top principal component of the return correlation matrix; high signals one dominant factor.

**Effective rank (`eff_rank`):** $\exp(H)$ of the normalized eigenvalue-spectrum entropy $H$; how many factors drive the cross-section. High is diversified, low is concentrated (the inverse view of PC1 share).

**Diversification recovery:** effective rank rising, and correlation concentration falling, after stress.

**Dispersion:** spread of returns across assets on a day.

**Breadth:** fraction of assets with positive returns (a direction measure).

**Left tail (p5):** 5th percentile of the cross-sectional return distribution.

**Drawdown:** decline from a trailing peak (63d SPX).

**VIX:** option-implied expected near-term S&P 500 volatility.

**HY spread:** high-yield corporate bond yield premium over Treasuries; widens in credit stress.

**Average dollar volume (ADV):** price $\times$ volume; liquidity proxy used for universe selection.

