# OpenReview forum: "Joint-Embedding Predictive Learning of Latent Market States in U.S. Equities"
_ICML.cc/2026/Conference — ICML 2026 regular_

### Official Review · Reviewer_QSzA · 2026-02-20

**Soundness:** 3
**Presentation:** 2
**Significance:** 3
**Originality:** 3
**Overall Recommendation:** 3
**Confidence:** 4

**Summary:**

This paper proposes a self-supervised learning framework based on JEPA, which treats daily equity cross-sections as unordered sets and compresses them into fixed factor tokens. By performing masked prediction within a latent space, the model filters out high-noise, hard-to-predict directional price or return information while precisely extracting persistent risk geometry features, providing a high-purity continuous state representation for identifying market regime evolution and predicting systemic recovery.

**Compliance With Llm Reviewing Policy:**

Affirmed.

**Key Questions For Authors:**

1 The paper claims that it can predict the evolution of the market’s latent geometric features and overall risk trends, including market recovery and the end of upward trends, and provide general directional guidance. However, the paper does not provide predictive comparisons with subsequent portfolio performance or overall market movements to demonstrate the accuracy and advantages of its predictions.

2 The matching between abnormal fluctuations of variables in the latent space and news events is somewhat far-fetched and overly subjective.

3 It is suggested to discuss the relationship between the proposed conceptual method and anomaly detection in stock market dynamics.

4 The introduction of the baseline methods is unclear, which affects the readability of the paper.

5 The presentation of the paper is poor. Many variables are not defined, and the expressions of formulas and variables are particularly imprecise. For example, the variables in Table 1 are not defined. In particular, in the core formula Equation (3), the function $sg$ is not defined, which seriously affects readability.

6 The paper does not provide a code link, and I am unable to understand these missing parts from the code.

**Limitations:**

Yes

**Strengths And Weaknesses:**

Strengths:
1 Using Joint-Embedding Predictive Learning to study stock market dynamics is a novel and interesting perspective.

2 The paper does not focus on predicting returns or prices, but instead concentrates on the overall shape and geometric characteristics of the market. This has meaningful implications for modeling financial markets with low signal-to-noise ratios.

Limitations:

1 The paper claims that it can predict the evolution of the market’s latent geometric features and overall risk trends, including market recovery and the end of upward trends, and provide general directional guidance. However, the paper does not provide predictive comparisons with subsequent portfolio performance or overall market movements to demonstrate the accuracy and advantages of its predictions.

2 The matching between abnormal fluctuations of variables in the latent space and news events is somewhat far-fetched and overly subjective.

3 It is suggested to discuss the relationship between the proposed conceptual method and anomaly detection in stock market dynamics.

4 The introduction of the baseline methods is unclear, which affects the readability of the paper.

5 The presentation of the paper is poor. Many variables are not defined, and the expressions of formulas and variables are particularly imprecise. For example, the variables in Table 1 are not defined. In particular, in the core formula Equation (3), the function $sg$ is not defined, which seriously affects readability.

6 The paper does not provide a code link, and I am unable to understand these missing parts from the code.

---

> ### Author Rebuttal · Authors · 2026-03-31
>
> ### **Does not provide predictive comparisons with subsequent portfolio performance or overall market movements to demonstrate the accuracy and advantages of its predictions.**
> We respectfully remind the reviewer that this is not a return-prediction or portfolio-optimization paper, and such comparisons are out of scope. The paper addresses this explicitly in various places: Section 1 ("**This is a representation learning study**. We do not optimize for expected returns or claim implementable trading strategies") and the Impact Statement ("we make no claims that the embeddings are suitable for trading, price prediction, return forecasting, market timing, or alpha generation"). We also report that the representation drops directional information: kNN retrieval shows weak association with market returns and breadth. That is an intended property of the JEPA objective, since the goal is persistent risk geometry. The paper does not claim to predict market movements, and the evaluations are deliberately designed to characterize the structural content of the learned representation, not to test trading performance.
>
>
> ### **The matching between abnormal fluctuations of variables in the latent space and news events is somewhat far-fetched and overly subjective.**
> We agree that the top-3 displacement examples and the cluster-level news labels are illustrative and somewhat subjective; they are meant only to give intuition for the latent space, not to serve as the main evidence. The quantitative claims instead rest on formal tests: the displacement enrichment analysis in Sec. 3.4 (OR = 32.3, one-sided Fisher p < 10^-4 for stress days) and the retrieval metric Δ(W) with shuffled nulls and bootstrap CIs.
>
>
> ### **It is suggested to discuss the relationship between the proposed conceptual method and anomaly detection in stock market dynamics.**
> We thank the reviewer for this suggestion. Section 3.4 is closely related to anomaly detection, though more precisely the displacement metric `delta_t = ||z_t - z_{t-1}||_2` acts as an unsupervised change-point score in learned state space. It is strongly enriched on realized stress days (OR = 32.3), however, not on scheduled announcements (OR = 1.3; one-sided Fisher p = 0.47), consistent with retrospective anomaly detection rather than event forecasting. In rebuttal we further benchmarked it against standard risk signals and found it is competitive with the best individual signals on stress enrichment (see our response to u7Yx).
>
>
> ### **Paper presentation / introduction of the baseline methods is unclear**
> We appreciate the reviewer raising this and recognize this as a real concern. The paper uses different baseline families per evaluation, which may have created the impression that baselines were missing when each section has its own controlled comparison (time/volatility/spectrum for geometry alignment; HAR and handcrafted descriptors for transition prediction; shuffled/seasonality/risk-summary controls for news retrieval; JEPA vs MAE and masking ablations for training). We agree that consolidating definitions would help. For reference:
> - Eq. (3): `p_phi` maps visible-token encodings (plus mask tokens and positional embeddings) to predictions at masked positions.
> - `SAB` = self-attention block; `ISAB` = inducing-point self-attention; `PMA` = pooling by multihead attention (all from the Set Transformer framework).
> - Table 1: `+/- 21d exclusion` = neighbors within 21 trading days removed; `gap >= 63d` = only day-pairs 63+ days apart; `OR` = odds ratio; `Delta(W)` = news-similarity gain at exclusion window W.
>
> These are the needed clarifications at first use.
>
>
> ### **The paper does not provide a code link, and I am unable to understand these missing parts from the code.**
> The implementation (including the Set Transformer tokenizer and JEPA architecture) was included as a runnable zip file in the supplementary material.  We agree that a clearer pointer in the main text would have helped.

---

> > ### Author Rebuttal · Reviewer_QSzA · 2026-04-05
> >
> > I maintain my scores.

---

> > > ### Author Response · Authors · 2026-04-08
> > >
> > > We thank the reviewer for their time and consideration.

---

### Official Review · Reviewer_u7Yx · 2026-02-27

**Soundness:** 3
**Presentation:** 3
**Significance:** 3
**Originality:** 3
**Overall Recommendation:** 4
**Confidence:** 4

**Summary:**

The paper studies how to represent the U.S. stock market using a Joint Embedding Predictive Architecture (JEPA). Its main contribution is a self-supervised framework that aims to capture the market’s risk geometry, rather than just predicting returns or direction. By combining a permutation-invariant tokenizer with a temporal masking task, the authors compress high-dimensional daily cross-sectional stock data into compact market state embeddings. The paper also proposes a comprehensive evaluation setup. Using linear probes, the authors show that the embeddings retain key geometric features such as volatility and correlations. With k-nearest-neighbor (kNN) search, they further demonstrate that nearby embeddings correspond to similar market states. Finally, they compare latent state transitions with external news topics.

**Compliance With Llm Reviewing Policy:**

Affirmed.

**Final Justification:**

Although the method and the research question in this paper are not novel, it is still a meaningful attempt. I previously tried an MAE-based approach to study this problem (how to effectively represent market states), but the results were mediocre; this paper introduces JEPA to address it. In the initial version, the experiments had many issues, but the authors added revisions during the rebuttal, which addressed my concerns. The value of this work lies in providing a solid foundation for other researchers through detailed experiments; compared with work that merely stacks models to chase SOTA, I appreciate this type of work more.

**Key Questions For Authors:**

1. Could the authors provide more informative and convincing experimental analyses? If so, I would be willing to reconsider and potentially raise my score. Although the model design and training strategy in this paper follow existing, well-established paradigms, I still appreciate the attempt to apply them to a financial setting. My main concern is that, despite the large number of experiments, the current results do not yet seem strong enough to clearly demonstrate the effectiveness and importance of the proposed design choices. At a more fundamental level, most of the evaluation metrics in the paper mainly test the correlation or alignment between the learned embeddings and traditional market structure measures (such as volatility, correlation structure, and geometric descriptors). These results do show that the embeddings capture some structural information, but they mostly demonstrate that the embeddings resemble existing measures, rather than that the embeddings are better than or go beyond them. Thus, the methodological contribution remains unclear on the key question of whether the embeddings offer new information beyond traditional indicators.

**Limitations:**

yes

**Strengths And Weaknesses:**

Strengths:
1. The paper addresses a core challenge in financial modeling by introducing a permutation-invariant cross-sectional tokenizer. This allows the model to handle a changing universe of stocks and the lack of any natural ordering. As a result, heterogeneous and unordered daily asset features are mapped into fixed-dimensional latent factor tokens, leading to more robust and stable representations.
2. The paper explores the use of JEPA for financial time series modeling, which offers a new way to learn non-stationary market states. Instead of using a generative objective like MAE, the model performs masked prediction directly in embedding space. This design encourages the extraction of temporally consistent signals from noisy price movements.
3. The study employs a thorough experimental design to conduct a comprehensive evaluation of the learned embedding representations from multiple perspectives.

Weaknesses：
1. In the neighborhood semantics (3.2) test, kNN neighbors are similar in volatility and correlation, but not in market direction. The authors say this means the model learns risk structure, not direction. I do not think this is fully supported. Volatility and correlation are more persistent and easier to predict, while returns are noisy, so a masked prediction objective will naturally focus on them. Also, the inputs already include several explicit volatility features (realized, EWMA, etc.). The kNN result may just show the model compresses and mixes these inputs, not that JEPA learns a new structural representation. It would help to rerun the test after removing explicit volatility features.
2. In the geometry fidelity (3.3.1) test, the distance correlation is higher for the volatility-only baseline (0.44) than for the JEPA embedding (0.35). This suggests the embedding geometry may mostly reflect volatility regimes, not an independent abstraction of correlation structure. If a simple volatility proxy already explains much of the correlation-geometry distances, it is hard to see what JEPA adds on this metric. Overall, this result seems to show that the correlation structure is strongly driven by volatility, rather than clearly showing that JEPA learns structure beyond volatility.
3. In the predictive utility (3.3.2) test, the authors compare a logistic regression using only a few geometry scalars with another model that also adds the JEPA embeddings, and they claim the embeddings add extra predictive power. But this is not a balanced comparison. The geometry scalars are a small set of low-dimensional features, while the JEPA embeddings come from full cross-sectional stock data over long windows and are learned with a neural self-supervised method. They contain much more information, so it is not surprising they help. To show what is special about JEPA, the paper should compare it to other representations with similar information and size, or to strong baselines built from the same inputs.
4. In the stress indicator (3.4) experiment, the analysis does not include a systematic comparison with standard risk measures (e.g., realized volatility, VIX-type indices, drawdowns, or liquidity metrics). It is therefore unclear whether the displacement measure adds meaningful information beyond these indicators. If market stress and regime shifts are already well captured by standard measures, embedding displacement may just be a more complex, non-linear version of known risk factors rather than a new source of insight. The authors should run formal tests to show that displacement adds unique, non-redundant information beyond traditional risk benchmarks.
5. In the cross-modal alignment (3.5) experiment, the paper says it also compares against a volatility-only representation in the same retrieval setup, but I could not find those results in the main text or appendix. If they are included, please clearly point to the exact location. More broadly, volatility is a major market driver and is often linked to macro regimes, risk appetite, and event cycles, which can also shape news topics. So any observed alignment between embeddings and news topics may reflect a shared volatility regime, not that the model learns higher-level semantic or structural information.
6. In the ablation study, JEPA shows much higher correlation than MAE on the structural description metrics. A likely reason is the objective: JEPA aligns in latent space and does not need to reconstruct every raw input dimension, so it can downweight low-signal variables like directional returns. MAE must reconstruct all dimensions, including noisy return components, which can add gradient noise and make it harder to learn stable patterns such as volatility and correlations. So JEPA’s gain on this metric may come more from focusing on what is easy to predict than from inherently better representations. More controlled comparisons are needed to support a stronger claim.

---

> ### Author Rebuttal · Authors · 2026-03-31
>
> ### **Volatility dependence and no-vol retrain (3.2, 3.3.1)**
>
> We retrained the full model after removing the explicit volatility channels listed in the paper: `rvol_10`, `rvol_21`, `rvol_63`, `rvol_ratio`, `ewma_vol_hl10`, `ewma_vol_hl20`, `mkt_rvol_21`, and `rel_vol`. The signal weakens but clearly persists (`Mantel rho: 0.348 -> 0.321`, mean geometry `R^2: 0.408 -> 0.378`, recovery AUC gain: `0.243 -> 0.138`).
>
> The vol-only baseline remains stronger on the Mantel metric (`0.44` vs `0.35`), so we do not present 3.3.1 alone as a beyond-volatility result. The point of the no-vol retrain is narrower and important: it argues against a simple passthrough explanation, because after removing those explicit channels the embedding will still retain substantial geometry and recovery signal.
>
>
> ### **Stronger baselines for predictive utility (3.3.2)**
>
> The reviewer asked for comparisons against strong baselines built from the same inputs, and against other representations with similar information and capacity. We added both.
>
> For strong traditional baselines: HAR, a Gaussian hidden Markov model, and a six-descriptor handcrafted market-state baseline (`logRV21`, `dispersion`, `breadth`, `left_tail`, `pc1_share`, `eff_rank`, each with daily/weekly/monthly persistence summaries, 18 features total). For the capacity-matched learned comparator: a MAE model with the same frozen tokenizer, same encoder architecture, same data, and same split, differing only in the training objective.
>
> | Task | Baseline | + JEPA | + MAE | JEPA `Delta` | MAE `Delta` |
> |---|---:|---:|---:|---:|---:|
> | `recovery_eff_rank_UP` | `0.690 / 0.139` | `0.800 / 0.324` | `0.723 / 0.200` | `+0.110 / +0.185` | `+0.033 / +0.061` |
> | `stress_corr_mean_UP` | `0.549 / 0.085` | `0.671 / 0.093` | `0.541 / 0.067` | `+0.122 / +0.008` | `-0.008 / -0.018` |
> | `recovery_corr_mean_DOWN` | `0.969 / 0.818` | `0.960 / 0.683` | `0.973 / 0.887` | `-0.008 / -0.136` | `+0.004 / +0.069` |
> | `recovery_pc1_share_DOWN` | `0.972 / 0.837` | `0.754 / 0.360` | `0.937 / 0.770` | `-0.218 / -0.476` | `-0.035 / -0.067` |
>
> Under these stronger controls, the clearest additive value survives on `recovery_eff_rank_UP` (`0.690 / 0.139 -> 0.800 / 0.324`), and `stress_corr_mean_UP` also improves mainly in AUC. MAE is weaker overall. The broader picture is selective rather than uniform, so our conclusion is this: JEPA adds complementary signal on slow diversification recovery from raw inputs, rather than broad dominance over the entire six-descriptor handcrafted baseline.
>
>
> ### **Displacement versus standard risk measures (3.4)**
>
> We added the missing benchmark against standard risk measures on the same held-out split (train `<= 2022`, val `2023`, test `2024-2025`).
>
> - Benchmark signals: `|SPX return|`, 5d rolling SPX vol, 63d SPX drawdown, VIX level, `|VIX change|`, HY spread level/change, and a liquidity proxy (cross-sectional median log dollar volume).
> - On top-1% Crash+VIX enrichment, JEPA displacement is competitive with the best individual signals: JEPA `14.96x`, `|SPX return|` `14.96x`, VIX `13.30x`, drawdown `11.63x`, liquidity `4.99x`.
> - For the incremental test (full risk bundle alone versus bundle plus JEPA displacement), Crash+VIX scores `0.985/0.586` AUC/AP vs `0.983/0.595`; All Systematic `0.716/0.298` vs `0.715/0.303`. Paired bootstrap intervals for both deltas include zero.
>
> As a standalone signal, JEPA displacement is competitive with the strongest stress indicators. When added to the full traditional risk bundle, the held-out gains are small and the bootstrap CIs cover zero. Thus, we interpret displacement as a useful label-free stress summary, rather than as a clearly independent signal beyond that bundle.
>
> ### **News retrieval controls (3.5)**
> We ran the retrieval evaluation against explicit baselines. The latent exceeds vol-only at `W >= 5` (`Delta(W=63) = 0.0053` v `0.0024`), and the no vol retrained model is comparable or stronger (`Delta(W=63) = 0.0080`), so the signal is not reducible to the explicit volatility channels. These results support a genuine alignment signal tied to risk geometry. A compact vol+correlation baseline is also equal or strong at many windows, so we view the news result as supporting evidence for the learned representation rather than the sole basis for a uniqueness claim.
>
> ### **JEPA v MAE**
> This is a very sharp point and we agree with the reviewer's mechanism. For a controlled comparison, we trained a matched MAE with the same tokenizer, encoder, data, and split, and evaluated both atop the same six-descriptor baseline. JEPA outperforms MAE on all six structure descriptors, and on held-out `recovery_eff_rank_UP` its additive gain is about 3x MAE's, while MAE's gain is not statistically supported. This is the pattern expected if latent prediction is a better fit than reconstruction for persistent risk geometry, and is is part of the motivation for this contribution.

---

> > ### Author Rebuttal · Reviewer_u7Yx · 2026-04-02
> >
> > Thank you for your response. I will adjust the score to 4.

---

> > > ### Author Response · Authors · 2026-04-03
> > >
> > > We sincerely thank the reviewer for their constructive and insightful feedback, which has helped clarify several aspects of our work.

---

### Official Review · Reviewer_KHSw · 2026-03-12

**Soundness:** 2
**Presentation:** 2
**Significance:** 2
**Originality:** 2
**Overall Recommendation:** 4
**Confidence:** 3

**Summary:**

This paper presents a self-supervised representation learning framework for U.S. equity markets, combining a permutation-invariant Set Transformer tokenizer with a Joint-Embedding Predictive Architecture (JEPA) temporal encoder. The authors report that the embeddings encode realized volatility and correlation geometry strongly, but encode market direction weakly. An ablation against a Masked Auto-Encoder (MAE) baseline shows JEPA consistently outperforms MAE on structure descriptors.

**Compliance With Llm Reviewing Policy:**

Affirmed.

**Final Justification:**

Based on the current results, the authors have largely addressed my concerns. If these promised changes can faithfully go into the final paper, I am raising my score to 4.

**Key Questions For Authors:**

1. Can you provide any analysis of what each factor token captures? For instance, do the 24 seed vectors in the PMA attend preferentially to equities with specific sector, size, or momentum characteristics? Do specific factor slots consistently activate during particular market regimes (e.g., crisis, recovery)?
2. Could you provide a theoretical or empirical justification for why the standard JEPA objective is well-suited for this domain? For example, how does the model's architecture or loss function address the challenges posed by heavy-tailed, non-stationary financial returns?
3. The paper's introduction acknowledges a rich literature on modeling market states from financial econometrics (e.g., DCC-GARCH, Markov Switching Models). However, these models are absent from the empirical evaluation. Could you elaborate on the rationale for this omission? Including a comparison, especially for the geometry transition prediction task, against a representative econometric model would be critical for contextualizing the performance of your approach and substantiating its contribution.

**Limitations:**

yes

**Strengths And Weaknesses:**

Strengths
* Clear research question: The paper learns a compact market-state embedding that captures second-moment geometry while discarding direction, and then rigorously characterizes what the embedding encodes. The explicit disclaimer that this is a representation learning study, not a trading strategy, is appreciated.
* Diverse evaluation methods: Rather than relying on a single metric, the authors design seven complementary evaluations, which collectively triangulate the properties of the learned representation. The temporal purging strategy shows care in preventing leakage.
* Honest limitations: The paper openly acknowledges the asymmetry in predictive utility, the dominance of the volatility baseline in geometry alignment, the lag-0 nature of displacement detection, and liquidity selection bias. This intellectual honesty is commendable.

Weakness
* Originality: The core architecture (JEPA with a Set Transformer front-end) is transplanted from video understanding (V-JEPA, I-JEPA) to financial time series with minimal domain-specific modification. The authors do not explain why this transfer is non-trivial—financial return series exhibit fundamentally different statistical properties from video frames: pronounced non-stationarity, strong serial autocorrelation with heavy tails, and multiplicative rather than additive noise structures.
* Comparison: The paper's single ablation compares JEPA against MAE—another method originating from computer vision. Given that the introduction extensively cites GARCH, DCC, HAR, and Markov Switching Models as the relevant prior literature, the complete absence of any quantitative comparison against these methods in the empirical evaluation is a critical gap.
* Data Adaptation: Financial return cross-sections are non-stationary, exhibit fat tails, volatility clustering, and significant microstructure noise. The paper applies standard per-day cross-sectional z-scoring and temporal z-scoring from training data, but takes no steps to address them.

---

> ### Author Rebuttal · Authors · 2026-03-31
>
> ### **Domain adaptation and why the JEPA objective fits**
> We truly appreciate this point. First, daily equity cross-sections mix persistent structure (volatility clustering, correlation regimes, concentration) with unpredictable noise (idiosyncratic returns, direction). This mix is the fundamental reason that latent prediction fit: compression, dropping what is not predictable across time gaps, rather than forcing reconstruction of every masked variable including the noisy ones.
>
> Regarding the transfer from vision not being automatic--we agree. While the foundational building blocks originate in vision, there are key differences that must be, and are, accounted for.
> - There is no spatial grid or strict locality, so the tokenizer must be permutation-invariant.
> - The universe changes daily as stocks list, delist, or cross liquidity thresholds; validity masks avoid survivorship bias by attending only to stocks actually present.
> - Returns are heavy-tailed and non-stationary, so normalization is train-only over time and robust within each day (median/MAD).
> - Temporal autocorrelation means naive masking leaks, so structured masking forces prediction across real time separation.
>
> Together these address the specific properties the review raises, non-stationarity, heavy tails, and serial autocorrelation, by forcing the model to learn structure that persists across temporal gaps rather than exploiting shortcuts available in gridded, stationary data.
>
> We additionally provide the JEPA/MAE ablation as empirical support. Architecture, tokenizer, masking, and data are held fixed. JEPA outperforms MAE in this domain, with the largest gains on PC1 share and effective rank. Both models are weak in direction. That is the expected signature: if latent prediction is filtering noise, reconstruction must allocate capacity across all masked variables including noisier directional components, while latent prediction can discard them.
>
>
>
>
>
> ### **On comparisons with traditional baselines**
>
> We investigate three additional baselines on the geometry-transition task under the same split and protocol: HAR, a Gaussian hidden Markov model, and a six-descriptor handcrafted market-state baseline (`logRV21`, `dispersion`, `breadth`, `left_tail`, `pc1_share`, `eff_rank`).
> JEPA's clearest additive signal is on diversification recovery (`recovery_eff_rank_UP`: `0.690/0.139` to `0.801/0.323` ROC-AUC / average precision). On geometry fidelity, JEPA remains positively aligned after temporal controls (`Mantel rho = 0.348` at `gap >= 63d`), although that test alone does not isolate structure beyond volatility. The stress tasks are mixed, and the displacement result is strongest as a descriptive stress indicator rather than a source of broad incremental predictive value.
>
> **To be precise about the scope**: the paper is a representation learning study. The goal is to show that a self-supervised model recovers meaningful market structure from raw cross-sectional inputs, not to outperform hand-engineered descriptors. Where traditional descriptors score higher, that reflects their direct construction from the target quantities. Systematically outperforming these is outside this contribution’s scope/claim.
>
>
>
>
> ### **On Factor tokens**
> We analyzed PMA slot behavior using (i) attention enrichment over volatility, momentum, and liquidity/size proxies, (ii) per-slot probes, and (iii) regime-conditioned activation statistics.
>
> The submitted model exhibits consistent but distributed specialization, individual slots show statistically significant biases (e.g., toward high-volatility or low-momentum equities), and slot activations vary across regimes (e.g., elevated activity during high-volatility/high-concentration periods). However, this structure is shared across multiple slots rather than forming discrete one-slot/one-factor mappings. To test whether sharper specialization is possible, we conducted a controlled PMA follow-up with explicit slot-specific pressure (same tokenizer architecture). In that setting, certain slots (e.g., slot_23, slot_4) interestingly show stronger concentration on high-volatility (+0.323, +0.303) and low-momentum (+0.294, +0.271) equities. With our analysis, we concluded that the slots capture factor-relevant structure in a more distributed form than clean/discrete one-slot/one-factor semantics.

---

> > ### Author Rebuttal · Reviewer_KHSw · 2026-04-04
> >
> > Thank you for the detailed rebuttal. The JEPA vs. MAE ablation as empirical support for the latent prediction hypothesis is a reasonable argument. However, the authors' position that "systematically outperforming hand-engineered descriptors is outside this contribution's scope" is difficult to accept. As the paper's central claim is that JEPA learns "useful representations" of market states. If the representation is evaluated purely against other machine learning objectives (MAE) and not against the domain's established models, the meaning of "useful" is left undefined. Even if the conclusion is that JEPA provides complementary rather than superior value that would itself be a meaningful and honest scientific finding. My overall evaluation of the paper is maintained.

---

> > > ### Author Response · Authors · 2026-04-07
> > >
> > > We thank reviewer for the continued engagement and the productive discussion. After deliberation, we agree with the reviewer's framing.
> > >
> > > The experiments support this framing directly: under the added domain controls (HAR, Gaussian HMM, and the six-descriptor handcrafted baseline), JEPA provides complementary value beyond what those baselines capture rather than dominating them across every task, with the clearest gain on diversification recovery (recovery_eff_rank_UP). The asymmetry across tasks is itself the finding.
> > >
> > > **We will make this framing explicit in the final version.**

---

### Official Review · Reviewer_UhzW · 2026-03-18

**Soundness:** 2
**Presentation:** 2
**Significance:** 2
**Originality:** 2
**Overall Recommendation:** 3
**Confidence:** 1

**Summary:**

The paper presents a JEPA framework to learn the market-state embeddding from US equity cross-sections. The framework uses Set Transformer to convert equities into learned factor tokens. The paper claim that

**Compliance With Llm Reviewing Policy:**

Affirmed.

**Final Justification:**

I maintain my score. My main concern remains the quality of the presentation, which I believe requires substantial revision

**Key Questions For Authors:**

NA

**Limitations:**

See weakness part.

**Strengths And Weaknesses:**

Strengths:

 + The paper is well-motivated

Weaknesses:

+ The technical contribution of this paper is limited.
+ The presentation can be improved. For example, the notations (SAB, PMA, ISBA) for equation 1 are never introduced

---

> ### Author Rebuttal · Authors · 2026-03-31
>
> We respectfully disagree that the technical contribution is limited. The paper studies whether a latent prediction architecture developed for video can produce meaningful representations in a domain that is seemingly orthogonal: unordered, changing daily equity cross-sections with heavy tails, non-stationarity, and no spatial structure. That investigation requires concrete architectural choices (permutation-invariant tokenizer, validity masking, structured temporal masking) and an evaluation suite to characterize what the resulting representation does and does not encode. Both the adaptation and the evaluation are new; no prior work, as far as we know, applies JEPA-style objectives to financial cross-sections or provides this type of multi-angle representation audit (geometry alignment, retrieval, transition prediction, displacement, cross-modal validation). We position this as a foundation that others can build on, not as a claim of solved market prediction, this is out of scope of this paper.
>
> Regarding notation, SAB, ISAB, and PMA refer to components from the Set Transformer framework (Lee et al., 2019), but we agree they should have been briefly introduced. For clarity, SAB is a self-attention block over the set, ISAB uses inducing points for scalable attention, and PMA performs attention-based pooling into a fixed number of tokens. We would add short definitions at first use in revision.

---

> > ### Author Rebuttal · Reviewer_UhzW · 2026-04-02
> >
> > Thank you for your rebuttal.
> >
> > To clarify, I do not have expertise in finance, so I am unable to fully evaluate the domain-specific contributions of this work.
> >
> > However, in terms of its contribution to the machine learning community, I feel that the paper does not provide a **sufficiently self-contained presentation**. For a machine-learning-focused conference, it would be beneficial to include introductory context for the problem setup and relevant financial concepts, as the current version is difficult to follow for readers without a finance background.
> >
> > As for the real-world significance of the contributions, I believe this would be better assessed by domain experts in finance.

---

> > > ### Author Response · Authors · 2026-04-03
> > >
> > > Thank you for clarifying. We agree that the presentation can be improved for accessibility to a broader ML audience.
> > >
> > > At the same time, we view this primarily as an issue of exposition rather than with contribution. The current submission currently makes the ML setup explicit: Section 1 defines the problem as learning a day-level representation from unordered equity cross-sections using a JEPA objective, and explicitly scopes the work as representation learning. Section 3.3 then defines the specific geometry metrics used in evaluation (corr mean, PC1 share, effective rank).
> > >
> > > Some of the additional details asked for are also present in the submission, but deferred to the appendix. Table 7 provides the full per-equity feature definitions (28 features spanning returns, volatility, volume, and market-relative quantities), and Appendix B.1 defines the Set Transformer components used in Eq. 1.
> > >
> > > We agree, however, surfacing a brief high-level summary of these feature groups and key financial qualities in the main body would improve accessibility for readers without a finance background.
> > >
> > > In revision, we would address this directly by adding short inline definitions of SAB/ISAB/PMA and a brief main-text summary of the feature groups and financial terms, while keeping fuller implementation details in the appendix. We hope this clarifies the remaining concern and how it can be addressed through improved exposition.

---

### Decision · Program_Chairs · 2026-04-30

**Decision:**

Accept (regular)

**Comment:**

This paper received mixed reviews. The two negative reviews are either low-confidence or rest largely on misreadings of the submission. The two weak-accept reviews were more informative: they raised substantive concerns about volatility confounds, missing econometric baselines, and the absence of domain-specific adaptation, and the rebuttal exchange, and the subsequent rebuttal from the authors meaningfully clarified how the contribution should be framed (e.g., complementary to, rather than superior over, existing econometric tools). Both reviewers were satisfied by this reframing, though neither expressed enthusiasm. The paper also suffers from significant presentation issues flagged by all reviewers, which should be fixed in revision. With the revised framing and the committed presentation fixes incorporated, the paper could sit above the accept threshold.